# A dichotomy of gene regulatory associations during the activated B-cell to plasmablast transition

Mario Cocco[1],*, Matthew A Care[1,2],*, Amel Saadi[1], Muna Al-Maskari[1,3], Gina Doody[1], Reuben Tooze[1]

**The activated B-cell (ABC) to plasmablast transition encompasses the cusp of antibody-secreting cell (ASC) differentiation. We explore this transition with integrated analysis in human cells, focusing on changes that follow removal from CD40-mediated signals. Within hours of input signal loss, cell growth programs shift toward enhanced proliferation, accompanied by ER-stress response, and up-regulation of ASC features. Clustering of genomic occupancy for IRF4, BLIMP1, XBP1, and CTCF with histone marks identifies a dichotomy: XBP1 and IRF4 link to induced but not repressed gene modules in plasmablasts, whereas BLIMP1 links to modules of ABC genes that are repressed, but not to activated genes. Between ABC and plasmablast states, IRF4 shifts away from AP1/IRF composite elements while maintaining occupancy at IRF and ETS/IRF elements. This parallels the loss of *BATF* expression, which is identified as a potential BLIMP1 target. In plasmablasts, IRF4 acquires an association with CTCF, a feature maintained in plasma cell myeloma lines. Thus, shifting occupancy links IRF4 to both ABC and ASC gene expression, whereas BLIMP1 occupancy links to repression of the activation state.**

## Introduction

The differentiation of plasma cells (PCs) from B-cells depends on epigenetic reprogramming, which is driven by sequential shifts in transcription factor (TF) expression and is division linked (1, 2). In the context of B-cell differentiation, independent of the germinal centre, reaction of the intermediate steps consists of the activated B-cell (ABC) state, which may also be referred to as a pre-plasmablast (3), and the plasmablast, the immediate proliferating precursor of the quiescent PC (1). Although we understand many elements of these intermediate steps, including key transcriptional regulators and relationships to certain types of mature B-cell neoplasm, details of the ABC to plasmablast transition are limited.

An informative approach to analysis of cell state transitions is the application of time course gene expression data (4). This can be used to resolve the sequence of co-regulated gene expression in coordinately responding cells, as observed in PC differentiation (2). Analysis with bioinformatics networking tools allows the resolution of fine-grained patterns of temporal co-expression across such differentiating cell populations (5), which by inference may enrich for common gene regulatory input (6, 7). This can be tested against independently derived data such as the genomic occupancy patterns of key TFs at specific differentiation states encompassed in the expression time course. Although defining the combinatorial logic controlling the expression of individual genes is highly challenging (7), associations between TF occupancy and the expression of multiple genes in a co-regulated gene module can allow the identification of shared regulatory enrichment linked to the cell state transitions (8).

Three TFs—IRF4, BLIMP1, and XBP1—have been principally linked to the reprogramming of gene expression that underpins the ABC to plasmablast transition (1). Detailed models have emerged which place these factors along with input activating signals in a hierarchy wherein IRF4 sits downstream of cytokine and NFkB-driven signaling pathways, and BLIMP1 downstream of IRF4 (9, 10, 11). BLIMP1 in turn controls XBP1 expression potentially both through transcriptional de-repression, and through ER-stress/unfolded protein response (UPR)–related pathways. The former, for example, links BLIMP1 to XBP1 up-regulation via repression of PAX5 (12), and the latter links BLIMP1 to the control of immunoglobulin gene transcription and the alternative poly-adenylation switch from membrane-bound to secretory forms (13). XBP1 is linked to functional secretory optimization, although lack of XBP1 does not preclude phenotypic differentiation (14, 15, 16). Deletion of each of these TFs in a B-cell–specific fashion in mice provides evidence for this broad epistatic relationship. IRF4 deficiency arrests B-cell differentiation before the ABC/pre-plasmablast phase (9, 10). BLIMP1 deletion precludes plasmablast/PC differentiation but allows transition to the ABC/pre-plasmablast phase (11), whereas XBP1 deletion allows the generation of phenotypic PC-like populations which, however, lack optimization for the secretory function of the equivalent normal population (14).

At the level of DNA occupancy, IRF4 displays an important feature which is that its potential binding sites and hence contribution to gene regulation can be modified by partner TFs. IRF4 generally

[1]Division of Immunology and Haematology, Leeds Institute of Medical Research, University of Leeds, Leeds, UK   [2]Bioinformatics Group, Institute of Molecular and Cellular Biology, University of Leeds, Leeds, UK   [3]Department of Medicine, Sultan Qaboos University Hospital, Muscat, Oman

Correspondence: r.tooze@leeds.ac.uk
*Mario Cocco and Matthew A Care contributed equally to this work

binds DNA weakly in isolation unless expressed at high level and more commonly binds with cofactors. Important among these are the E26 transformation-specific (ETS) factors PU.1 and SPIB, which allow IRF4 binding at ETS/IRF composite elements (EICEs) and variations thereof (17, 18), and the AP1-factor basic leucine zipper transcription factor, activating transcription factor-like (BATF) which partners with IRF4 at AP1/IRF4 composite elements (AICEs) (19, 20). Thus, IRF4 has the capacity to bind different gene regulatory elements depending on patterns of co-factor expression. An elegant model of graded expression of IRF4 and changes in co-factors and occupancy patterns has been characterized in murine antibody-secreting cell (ASC) differentiation (21). In this model, IRF4 binding shifts from an AICE- and EICE-centred pattern during early B-cell activation toward a pattern focused on motifs of an interferon-sensitive response element (ISRE) type in ASCs. The latter reflecting a more generic mode of interferon regulatory factor binding as homo- or heterodimers, which in the proposed model is facilitated by higher levels of IRF4 expression. These changes may at least in part be regulated by BLIMP1, which, for example, is a repressor of the IRF4 co-factor SPIB (22). Furthermore, BLIMP1 and IRFs share overlapping DNA-motif preferences (23, 24, 25, 26); therefore, substantial potential for interplay exists between these factors.

Expression of BLIMP1 (encoded by the gene *PRDM1*) is a critical step in ASC differentiation (11). In the absence of BLIMP1 expression, components of the B-cell transcriptional program fail to be repressed, and the reprogramming of PCs for secretory activity is abortive, including incomplete metabolic reprogramming and failure to switch from membrane to secretory forms of immunoglobulin (13, 26). Such findings in murine genetic models have extended the contribution of BLIMP1 to include a more extensive role in positive regulation of induced gene expression. In humans, malignancy inactivation of BLIMP1 occurs in a subset of diffuse large B-cell lymphomas (DLBCLs) sharing many features with the physiological ABC state. In this context, loss of BLIMP1 function is predominantly interpreted as failure of BLIMP1-associated repressive functions, which in conjunction with other oncogenic events trap the malignant cells at the ABC to plasmablast transition (27, 28, 29). However, we know relatively little about the extent to which BLIMP1 couples to either positive or negative regulation of gene expression during the analogous differentiation of primary human B-cells from ABC to plasmablast.

BLIMP1 mediates its role as a transcriptional repressor through the recruitment of epigenetic regulators, which include histone deacetylases HDAC1/2/3, histone methyltransferases G9A (EHMT2) and EZH2, as well as the histone demethylase LSD1, and the protein arginine methyltransferase PRMT5 (26, 30, 31, 32, 33, 34, 35, 36). Among these, the combination of HDACs and histone methyltransferases provide the potential to convert the epigenetic state of target genes from an open to repressed chromatin state, with recruitment of G9A and EZH2 providing the capacity to mediate the establishment of repressive methylation marks at H3K9 and H3K27 residues (26, 33, 37, 38).

In vitro models allow the sequential tracking of transitions between cell states during PC differentiation. In human models, CD40L-based activation has provided a central platform for understanding PC differentiation (39). Here, we explore the gene regulatory changes that characterize the transition between ABC and plasmablast states after removal from CD40L signaling in such

a model (5, 40). We assess the relationship to genomic occupancy and epigenetic state linked to BLIMP1, IRF4, and XBP1. These data reveal a dichotomy in association of these core TFs with the pattern of gene expression change that characterizes the ABC to plasmablast transition.

# Results

## ABC encompass a growth state of B-cell differentiation

To explore the ABC to plasmablast transition, we initially performed a time-course gene expression experiment from total human peripheral blood B-cells and explored the data with parsimonious gene correlation network analysis (PGCNA), an efficient computational approach recently developed in our laboratory (5, 41). Cells were sampled at day 0 (resting B-cell); on day 3 after activation with CD40L, anti-BCR, and cytokines (ABC); and then at intervals of 3, 6, 12, 24, 48, and 72 h (plasmablast) after transition into conditions (continued IL2 and IL21 only) that support the ABC/plasmablast transition. The network representing gene expression change over this differentiation window comprised 20 modules (Figs 1A and S1 and Table S1; https://mcare.link/abctopb). Gene signature and ontology enrichment was used to assess biological functions associated with each module, which illustrated effective separation of known biological pathways between modules (Figs 1B and S2 and Table S2). Summary designations for each module were derived from these enrichments. Overlaying the expression z-scores from the differentiation series then allowed visualization of the transitions in gene expression across the network as differentiation progressed (Fig 1C and D, and interactive network resources).

Considering the temporal transitions between the three primary cell states, resting B-cells preferentially express modules enriched in genes characteristic of the B-lineage and chromatin regulators (M6 and M11), peptide chain elongation (M5), a subset of the secretory apparatus related to the Golgi and glycoprotein biosynthesis (M12) as well as genes linked to endosomal vesicles and quiescence signatures (M1). Modules shared between the resting B-cell and ABC state (M2 and M3) link features of the B-lineage to characteristic elements of the NFkB and cytokine response pathways. At the ABC state, the modules expressed preferentially in resting B-cells were repressed, whereas elements of modules M2 and M3 are retained. This was accompanied by enhanced expression of modules on the one hand characteristic of the signaling pathway inputs (M10) and three modules of genes related to cell growth and division with a dominant impact of MYC and E2F target gene expression (M4, M7, and M9). The separation of these three modules sharing common linkage to MYC and E2F targets reflects relative enrichment of distinct biological processes such as hallmark signatures of cell cycle G2/M checkpoint (M4 and M7), DNA repair and mRNA-related metabolic processes (M7), and noncoding RNA, ribosomal RNA, and telomerase/Cajal body RNA localization (M9). Thus, the ABC state was characterized by a dominant signature of MYC- and E2F-related growth programs and sustained evidence of activating input signals.

By contrast, the eventual plasmablast state saw a silencing of B-cell modules (M2, M3, and M11), signal input modules (M10 and components of M2), and cell growth–related modules (M4, M7, and

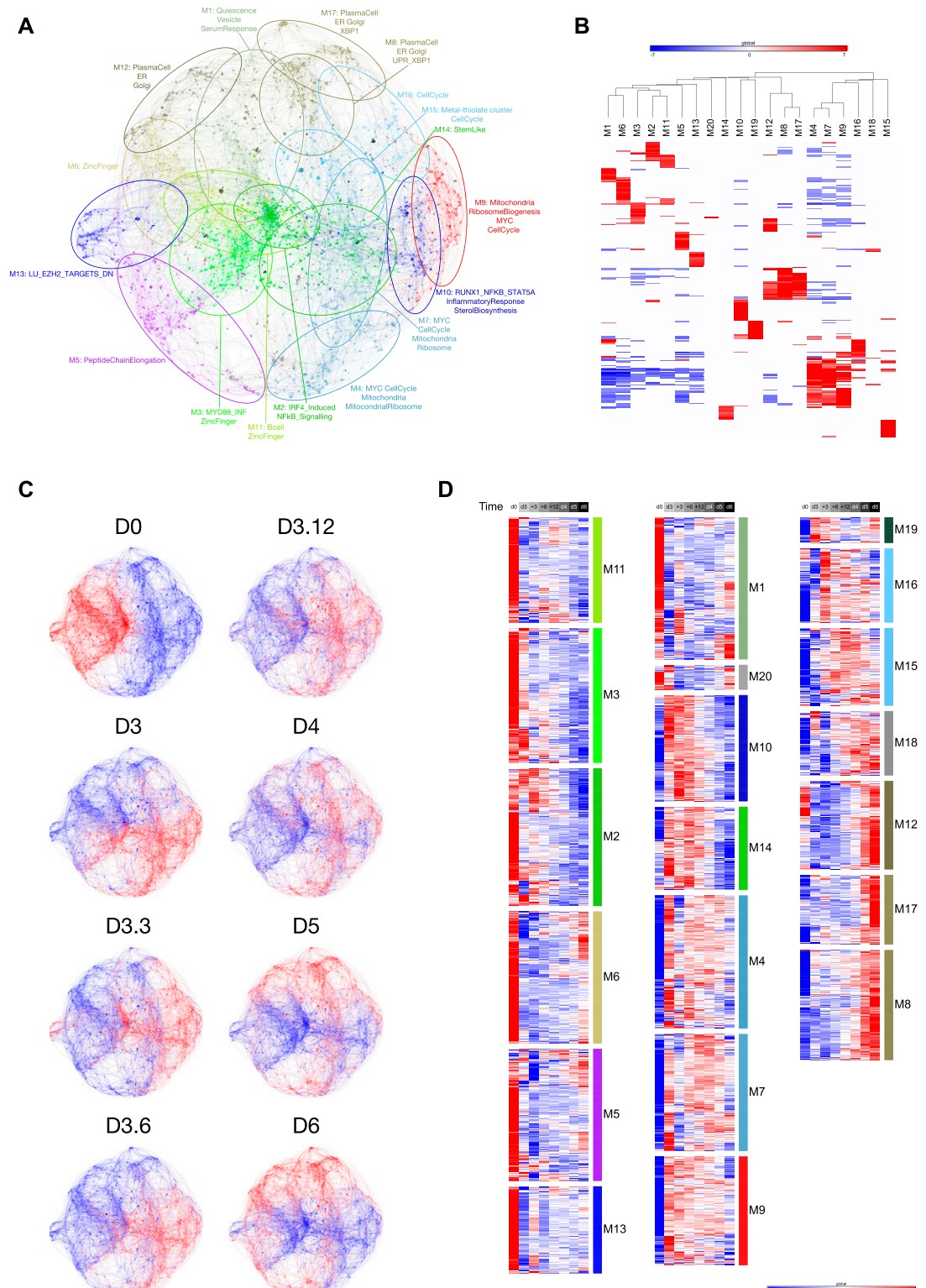

**Figure 1.   Application of parsimonious gene correlation network analysis to time course gene expression data of total peripheral blood B-cell differentiation to plasmablast state.**

**(A)** Network representation of the modular pattern of gene expression during the transition of B-cell to plasmablast. Module designations derived from gene ontology enrichment indicated with color code and ovals. Module genes are shown in Table S1, high-resolution version shown in Fig S1, and interactive version at https://mcare.link/abctopb. **(B)** Heat map summary representation of gene ontology and signature separation between network modules (filtered false discovery rate [FDR] < 0.05 and ≥5 and ≤1,000 genes; selecting the top 15 most significant signatures per module). Significant enrichment or depletion illustrated on red/blue scale, x-axis

M9) of the ABC state. These changes are paralleled by enhanced expression of modules related to the UPR and XBP1 signaling (M8), and the ER including additional XBP1 targets (M17). In addition, several modules expressed in the quiescent B-cell state are re-expressed in the plasmablast state, including genes related to the Golgi apparatus (M12), chromatin regulators including demethylases (M6), and the peptide chain elongation enriched module (M5). Thus, dominant growth programs are contained in the ABC state, whereas plasmablasts and resting B-cells share gene expression related to functional pathways.

### Transition from loss of signal input to cell cycle and secretory modules

Intermediate time points provide a more detailed view of the transitional states between signaling and growth programs and secretory functional programs that characterize the ABC to plasmablast transition (Fig 1C and D). The most immediate gene expression changes reflect the removal of CD40L and BCR signals that accompany transfer into renewed cytokine conditions. Within 3 h, down-modulation among specific signaling pathway genes was seen, including *TNFAIP3* and *RGS1* residing primarily in module M2 which is shared between the resting B-cell and ABC state (Fig 2A). Genes in the signal response module M10 such as *BATF* showed a slightly more delayed repression but decayed significantly from 6 h onward. These kinetics contrast with MYC and cell cycle–related gene expression modules of the ABC state (M4 and M7) including genes such as *KIF11*, *CENPK*, *BUB1B*, *CDK2*, *CCNA1*, and *MCM4* which were maintained to 48 h (Fig 2B and C). An additional module of cell cycle–linked genes (M16), particularly enriched for genes related to mitotic cell cycle, shows increased expression from 3 to 6 h and includes both the proliferation-related TFs *FOXM1* and *MYB* and the classical G1/S phase marker *MKI67* (Fig 2D). Consistent with the plasmablast population remaining in cell cycle, the expression levels of such proliferation-associated genes, while no longer at peak levels, remained considerably higher than in the initial quiescent B-cell state.

The central drivers of transcriptional reprogramming *IRF4*, *PRDM1* (encoding BLIMP1), and *XBP1* show similar patterns overall with some up-regulation at day 3 relative to day 0 and subsequent substantial increase over the following 72 h, with IRF4 showing the most rapid and XBP1 the most delayed kinetics of these factors (Fig 2E). Differentiation and initial secretory pathway gene expression is enriched in module M8 which includes XBP1 and PRDM1 along with UPR target genes which principally initiate expression from 12 to 24 h onward. Secretory program gene expression extends into modules enriched for distinct secretory pathway elements (M17 and M12), which include genes whose eventual peak expression is characteristic of the mature quiescent PC state (5).

These patterns of gene expression are consistent with the sequence of changes in cell phenotype, cell division kinetics, and transition to secretory activity seen across the differentiation. At a phenotypic level, there is a progressive transition to a CD20 low and CD38 high phenotype (Fig S3A and B), which becomes most pronounced after release from CD40L at day 3. This is also mirrored in a shift in CD30 expression which is a marker of the ABC state (3). This is maximal at day 3 and down-modulated by day 6 (Fig S3C). For cell division, a modest number of divisions occur over the first 3 d of activation as assessed by CFSE dilution, followed by a very rapid proliferative phase after release from CD40L (Fig S3A). At a functional level, ELIspot assays at day 6 confirmed extensive commitment to secretory activity with close to 50% of seeded cells recovered as ELIspot equivalents (Fig S3D). Thus, by day 6 of the differentiation, the phenotypic and functional characteristics of a predominant ASC population are established.

### Dynamics of gene expression for memory B-cells at the ABC to plasmablast transition

Memory B-cells provide a source of plasmablasts capable of generating long-lived PCs in vitro (40). We, therefore, next considered differentiation of memory B-cells in isolation, starting from the ABC state and using the same gene expression time course approach. The focused memory B-cell network comprised 21 modules that showed significant enriched biology (Figs 3A and B, S4, and S5 and Tables S3 and S4). These followed the three general patterns concordant with the total B-cell differentiation: early silencing, transient expression and late induction across the ABC to plasmablast transition (Fig 3C and D and https://mcare.link/abctopb). Modules reflecting the MYC-regulated growth program (m.M3), ribosome subunits and peptide chain elongation (m.M4), and the activation signal response (m.M5 and m.M8) dominated the ABC state. These were followed by transient up-regulation of cell cycle–related modules (m.M7 and m.M9), including the genes *MYB*, *MKI67*, and *FOXM1*, through the induction of secretory program components (m.M1 and m.M2). Across the time course, the largest variance was seen in genes belonging to m.M1, m.M5, and m.M8. Whereas some divergence in module composition was evident between the total and memory B-cell–related gene expression patterns, the overall progression in gene expression was highly similar with a transient wave of proliferative gene expression after the decay of input signals and onset of secretory program from 24 to 48 h after release into transitional conditions.

### BLIMP1 and IRF4 occupancy in memory-derived plasmablasts

To explore the relationship of TF occupancy patterns and gene regulation, we focused our analysis on the memory B-cell–derived plasmablast population performing chromatin immunoprecipitation

---

(signatures) and y-axis (modules). Hierarchical clustering according to gene signature enrichment. For high-resolution version and extended data, see Fig S2 and Table S2. **(C)** Overlay of gene expression z-scores for all genes in the network shown in blue (low) to red (high) z-score color scale. Day 0 (D0) provides the starting reference point for the sequential expression patterns observed at the subsequent time points indicated following decimal point for samples between D3 and D4. **(D)** Heat map displaying the pattern of gene expression across the time course module numbers indicated on the right, z-score gene expression blue (−1.8 low)–red (+1.8 high) color scale as indicated in the right lower edge, showing the median expression across three donors per time point. Modules divided into three broad categories of kinetics on at D0 going off, transient expression between D0 and D6, up-regulated at late time points.

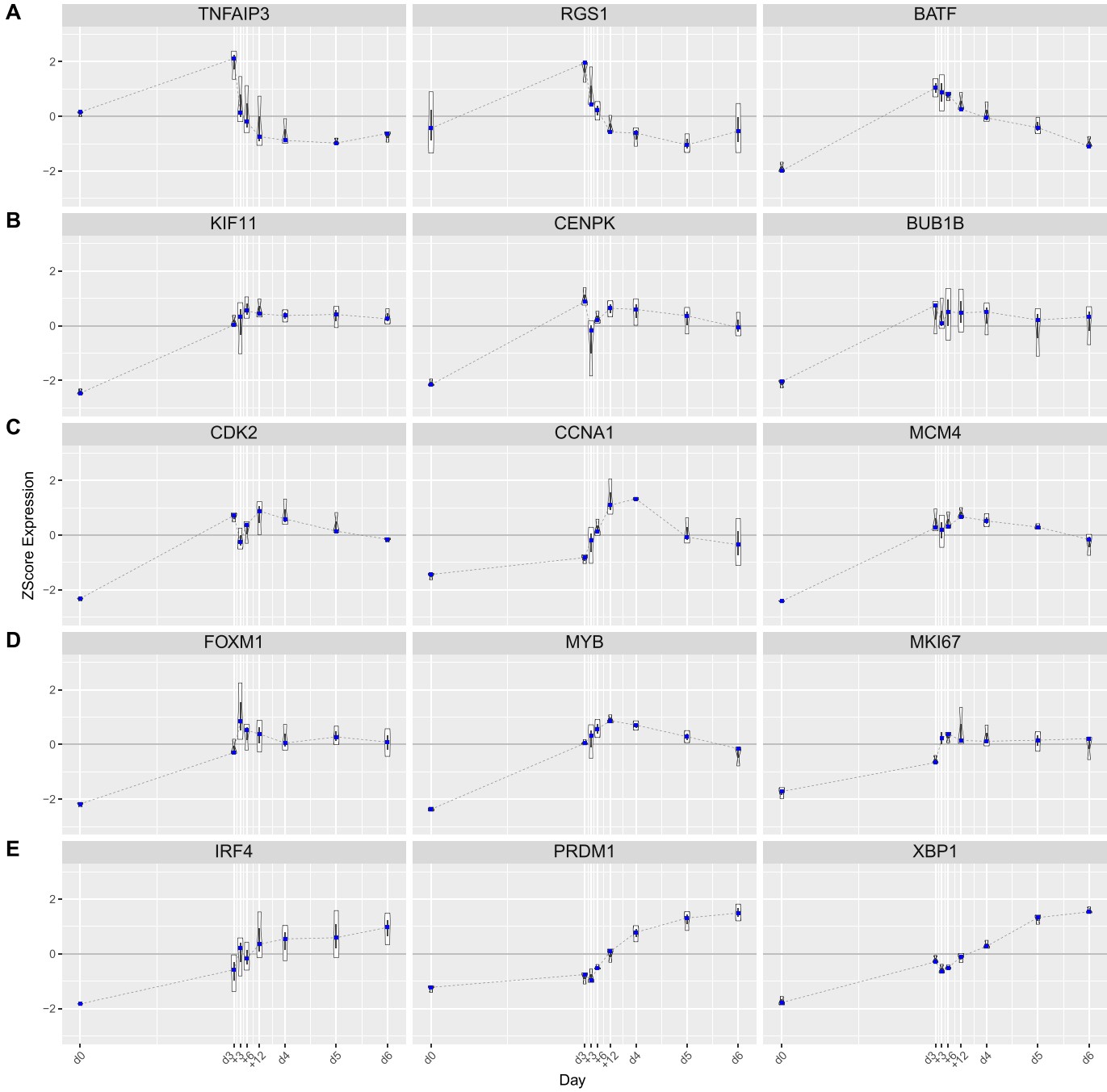

**Figure 2. Kinetics of exemplar genes.**
Violin plots of individual gene expression for selected genes showing expression z-score on the y-axis and time point in days and hours along the x-axis for the indicated genes above each graph. Violin plots display the distribution (n = 3 donors) along with median (blue square) and the inter-quartile range. **(A, B, C, D, E)** genes from M2 enriched for signaling response/immediate early genes, (B) M4, (C) M7, (D) M16 reflecting different patterns of cell cycle gene expression, and (E) core transcriptional regulators of the plasma cell state.

and sequencing (ChIP-seq) for BLIMP1 and IRF4. We identified 4,323 BLIMP1 occupancy sites of which the majority (69%) fell within intronic and intergenic regions and close to a quarter in promoter regions. For IRF4, we identified 9,512 peaks of which a greater proportion (44%) fell within promoter regions and just under half in inter- or intra-genic regions (Fig 4A and Table S5). Thus, IRF4

displayed a greater tendency for promoter occupancy. Of the total peak set, 1,717 regions were co-occupied by both factors (Fig 4B).

De novo motif detection of occupied sites for IRF4 and BLIMP1 independently confirmed the dominant enrichment of motifs matching the established consensus sequences (Figs 4C and S6A and B). IRF4-occupied sites were significantly enriched for motifs

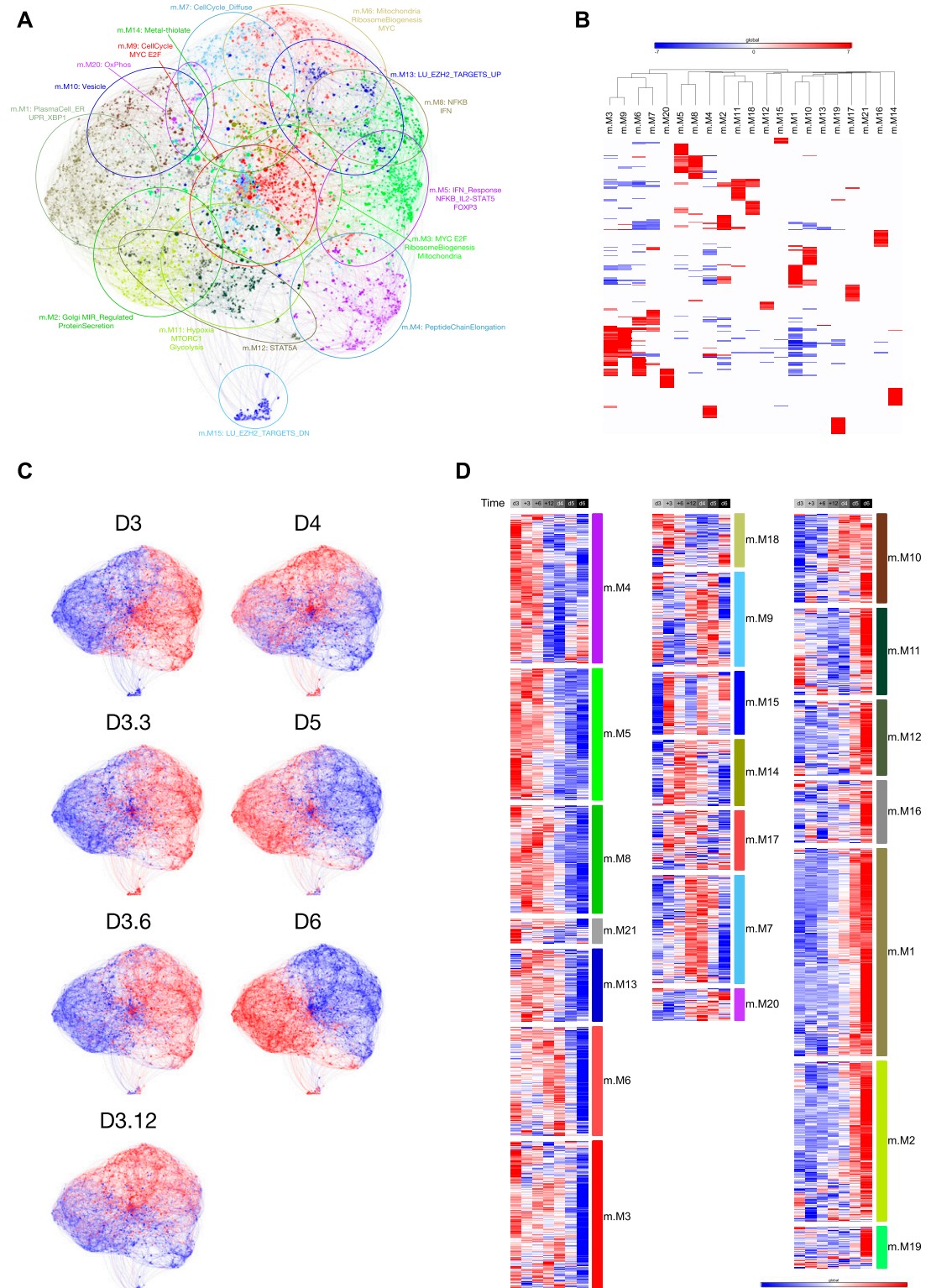

**Figure 3. Application of parsimonious gene correlation network analysis to time course gene expression data of memory B-cell differentiation from activated B-cell to plasmablast state.**
**(A)** Network representation of the modular pattern of gene expression during the transition of memory B-cell–derived activated B-cells to plasmablast state. Module designations derived from gene ontology enrichment indicated with color code and ovals. Module genes shown in Table S3, high-resolution version shown in Fig S4 and interactive version at https://mcare.link/abctopb. **(B)** Heat map summary representation of gene ontology and signature separation between network modules (filtered FDR < 0.05 and ≥5 and ≤1,000 genes; selecting the top 15 most significant signatures per module). Significant enrichment or depletion illustrated on red/blue

matching EICE and ISRE patterns, two of the primary modes of DNA binding associated with this factor. While IRF-binding motifs including the core GAAA consensus were also frequent and significantly enriched, AICE motifs in either of its two configurations were significantly less common. Indeed, enrichment of motifs for the chromatin-looping factor CTCF were more common and more significantly enriched. For BLIMP1, de novo motif detection returned the BLIMP1 consensus and variants overlapping with ISRE motifs. Notably, CTCF motifs were not identified by de novo detection among BLIMP1-bound sites.

The BLIMP1 and the IRF family share an evolutionarily conserved partial overlap in motif preference. We and others have previously shown that BLIMP1 and IRFs differ in their preference for the first position of the GTG triplet in the shared consensus, AAGTGAAAGT, where selection of a C rather than G disfavors BLIMP1 occupancy (23, 26, 42). This observation was supported in de novo motif analysis of individual or co-occupied sites. BLIMP1 alone selected strongly for the AAGTGAAAGT consensus. Co-occupied sites included both BLIMP1-favored and IRF-favored variants of EICE and ISRE, whereas sites occupied by IRF4 alone were enriched for IRF-favoring variants (Fig 4C). Notably recovery of a CTCF/BORIS motif was restricted to sites occupied by IRF4 alone (Figs 4C and S6C). BLIMP1 and IRF4, therefore, showed partially overlapping genomic occupancy in human plasmablasts, selecting for closely related binding motifs, with independent occupancy by each factor enriching for preferred variations and linked to differences in co-occurring secondary motifs.

### BLIMP1, IRF4, and XBP1 occupy distinct regulatory element clusters

To profile the epigenetic state associated with IRF4 and BLIMP1 occupancy and relate this to the additional elements of the transcriptional network controlling differentiation, we performed ChIP-seq for H3K4me3 (peaks n = 50,422), H3K27ac (peaks n = 28,141), CTCF (peaks n = 47,475), and XBP1 (peaks n = 605). For both CTCF and XBP1, we recovered the appropriate primary TF motif (Fig S6D and E). For CTCF, matches to known motifs and related variants were highly significantly enriched, and the most common secondary factor motifs were of E-box type. XBP1 ChIP-seq provided the most limited peak set, but de novo analysis returned a match to the previously defined human XBP1 DNA-binding motif G(C/A) CACGT as the most significantly enriched motif (43). At a subset of sites, a CCAAT box was also evident (Fig S6E) and together these comprise the composite ER stress response element (44).

We then used the union of genomic regions that were occupied by XBP1, IRF4, or BLIMP1 to assess recurring patterns of chromatin marks and CTCF occupancy (Fig 5). We used K-means to resolve six regulatory clusters among these peak regions. This encompassed

three regulatory clusters of highly active regions with either 5′ or 3′ skewing of histone marks, or symmetric distribution around the TF site (U.K1-3), consistent with the general observation of widespread heterogeneity of histone modifications around TF sites (45). These clusters were promoter biased and relatively enriched for IRF4 and XBP1 occupancy. The fourth regulatory cluster was distinctively associated with CTCF occupancy and enriched for IRF4 binding relative to BLIMP1 or XBP1 (U.K4). The remaining two regulatory clusters were linked to weak (U.K5) or absent (U.K6) active histone marks. BLIMP1 binding was relatively enriched at both these clusters but in particular for cluster U.K6, which was also relatively depleted of XBP1 and CTCF binding.

Repeating the analysis centering on occupancy by each TF independently reinforced these differential patterns (Figs S7–S9). XBP1 associated primarily with open active chromatin in promoter regions in the absence of BLIMP1 (X.K2-4) and with a small subset of CTCF enriched regions (3.3%; X.K1) (Fig S7). By contrast, BLIMP1 was preferentially enriched in relatively inactive chromatin fractions comprising 64% of peak regions, either alone (B.K6) or with IRF4 co-occupancy (B.K5) (Fig S8). Other patterns of BLIMP1 occupancy included a small fraction with CTCF (2.1%; B.K3), whereas 34% of BLIMP1 binding was linked to active chromatin in promoter-biased regions (B.K1 and 2) and sites with weaker active marks and bias to exonic and intronic regions (B.K4). IRF4 bridges these patterns binding primarily at active regions with promoter enrichment (47%; d6I.K1-4), as well as in relatively inactive chromatin enriched for intergenic and intronic regions and BLIMP1 occupancy (42%; d6I.K6) (Fig S9). IRF4 in the absence of BLIMP1 or XBP1 showed a distinct association with CTCF (11%; d6I.K5), confirming the results of de novo motif analysis, and contrasting with XBP1 and BLIMP1 which associated infrequently with CTCF. Thus, each of the core TFs of the plasmablast state is linked at the genome-wide level to a distinct pattern of epigenetic co-association.

### Modular gene expression link to TF occupancy and regulatory element clusters

To gain further insight into the link between regulatory clusters and expression patterns, we tested the association of the six regulatory clusters derived from the integrated and TF–specific analyses against the patterns of gene regulation observed in the memory B-cell PGCNA network. To do this, we focused on occupancy proximal to a gene promoter (10 kb no intervening TSS) as indicative of a potential regulatory interaction, looking for enrichment of such events across all genes in a module relative to background of all genes in the network. These analyses demonstrated that the regulatory element clusters defined using either integrated data or specifically for each TF were significantly and differentially linked to the modular patterns of gene expression defined by PGCNA, in a

scale, x-axis (signatures), and y-axis (modules). Hierarchical clustering according to gene signature enrichment. For high-resolution version and extended data, see Fig S5 and Table S4. **(C)** Overlay of gene expression z-scores for all genes in the network shown in blue (low) to red (high) z-score color scale. Day 3 (D3) provides the starting reference point for the sequential expression patterns observed at the subsequent time points indicated following decimal point for samples between D3 and D4. **(D)** Heat map displaying the pattern of gene expression across the time course module numbers indicated on the right, z-score gene expression blue (−1.8 low)–red (+1.8 high) color scale as indicated at the right lower edge, showing the median expression across three donors per time point. Modules divided into three broad categories of kinetics: (left) on at D3 going off, transient expression between D3 and D6, up-regulated at D6.

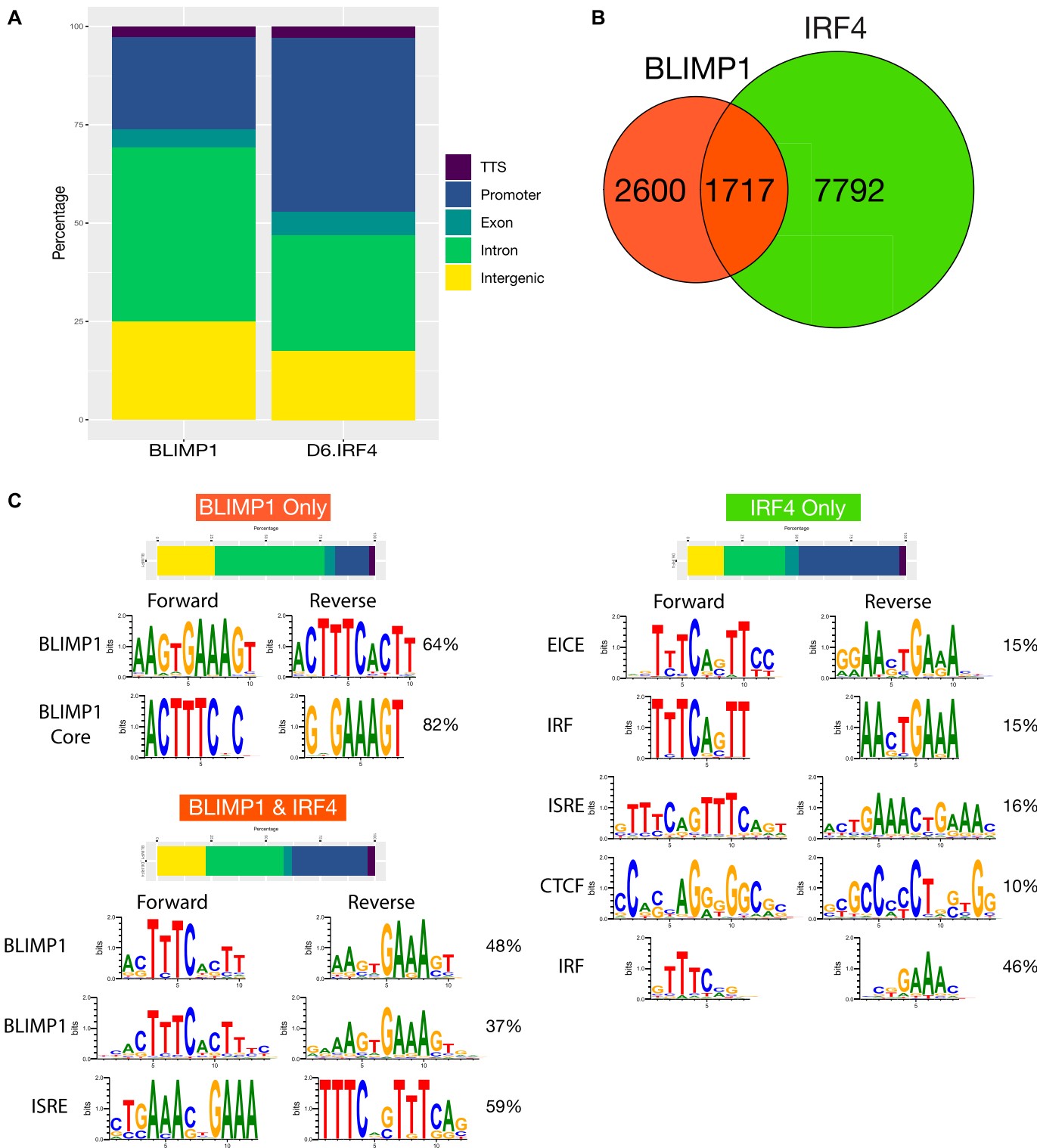

**Figure 4. IRF4 and BLIMP1 occupancy in human plasmablasts.**

**(A)** Relative distribution of BLIMP1 (left) and IRF4 (right) peaks identified in human memory B-cell derived plasmablasts divided according to genomic distribution, transcription termination site (TTS), promoter, exonic, intronic and intergenic as indicated in the color code to the right of the stacked bar graph (Promoter: −1 kb−100 bp, TTS: −100 bp−1 kb, Exonic/Intronic: > 100 bp from Promoter/TTS within gene, Intergenic: >1 kb from Promoter/TTS outside gene). ChIP data derive from individual samples for day 6. **(B)** Venn diagram depiction of BLIMP1 and IRF4 binding site overlap genome wide. **(C)** Relative genomic distribution and de novo motifs discovered at sites of BLIMP1-only, IRF4-only, and BLIMP1/IRF4 overlapping occupancy. **(A)** Shown is the genomic distribution as stacked bar graph color-coded as in (A) and the most significantly enriched motifs with percentage of peak regions with match to represented motif variant to the right. For each motif a summary designation is provided to the left, relating to a known motif match.

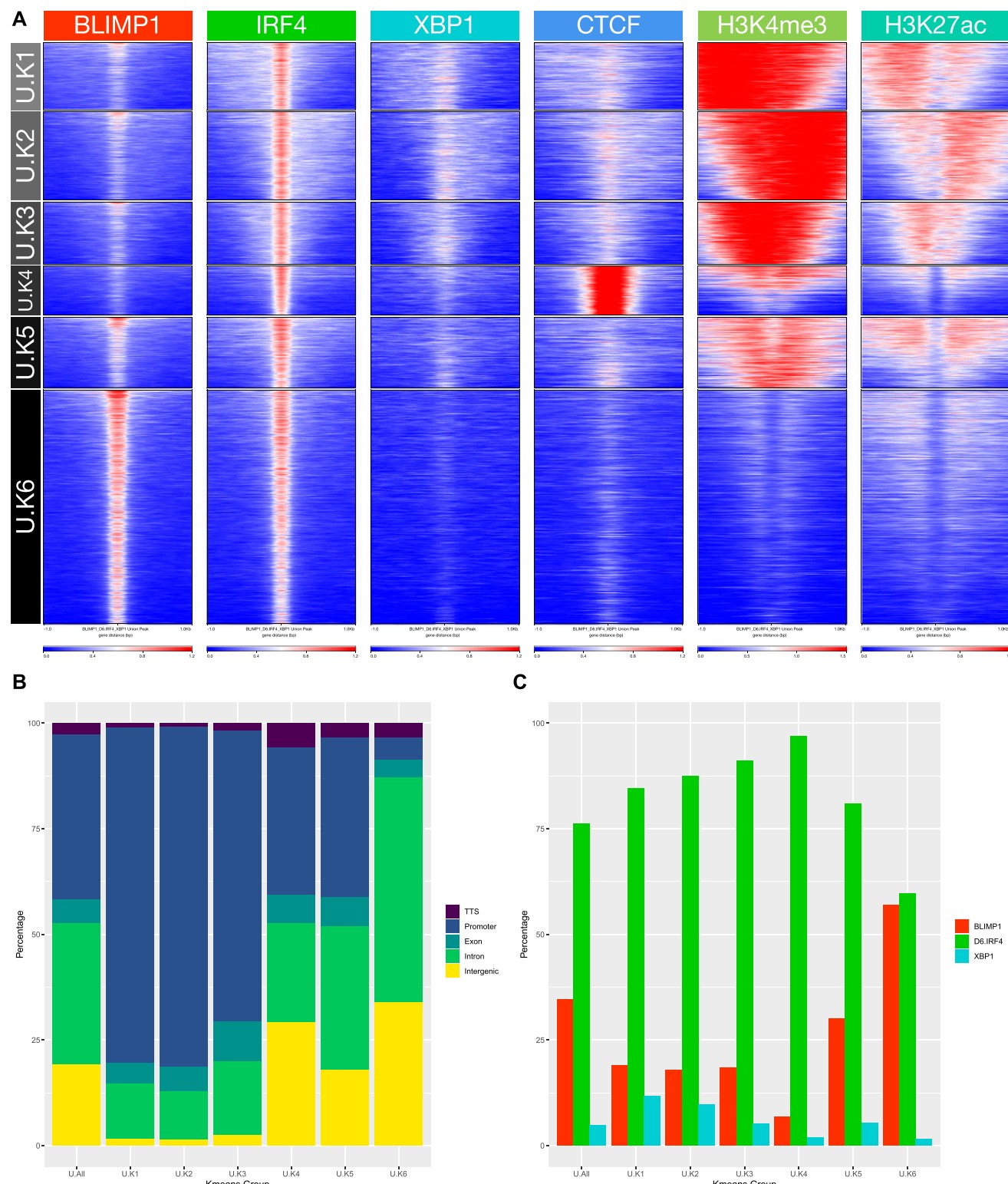

**Figure 5.  Epigenetic patterns associate with core TF occupancy at the plasmablast state.**
**(A)** deepTools heat map representation of K-means–clustered integrated ChIP-seq data from the plasmablast state. Data are clustered across the union of peaks for IRF4, BLIMP1, and XBP1 and encompassing data for CTCF, H3K4me3, and H3K27ac from equivalent cells. Six regulatory clusters are derived designated as U.K1-K6 on the left.
**(A, B)** Relative distribution of K-means clusters U.K1-K6 derived from (A) according to the genomic distribution, transcription termination site, promoter, exonic, intronic, and intergenic as indicated in the color code to the right of the stacked bar graph. **(A, C)** Percentage occupancy of individual TF binding across the K-means clusters derived (A) for each of the TFs indicated by the color code to the right of the figure (BLIMP1-red, IRF4-green, and XBP1-blue).

fashion which was concordant with known TF biology (Fig 6). Regulatory element clusters are significantly linked to the most characteristic and variant modules of gene expression across the network, including both induced and repressed genes. However, it is also evident that several gene expression modules that characterize the ABC/plasmablast transition lack positive association with IRF4, BLIMP1, or XBP1 regulatory element clusters. These, for example, include gene expression modules linked to oxphos (m.M20) and cell cycle, MYC, and E2F target genes (m.M9) or peptide chain elongation (m.M4). Such modules presumptively have dominant input from other transcriptional regulators, including MYC and E2F, during this differentiation window.

Considered from the point of view of gene expression, modules that were induced at the plasmablast stage and encompass core phenotypic and functional pathways of this state (m.M1, m.M2, m.M10, and m.M11) link to regulatory element clusters characterized by active marks, promoter enrichment, and IRF4 and/or XBP1 occupancy. These modules are neutral or anti-correlated (m.M1_PlasmaCell) with BLIMP1 occupancy. Reciprocally, modules characteristic of ABCs and repressed at the plasmablast state (m.M5 and m.M8) are correlated with regulatory element clusters linked to BLIMP1 occupancy and weak or absent active histone marks. Such modules of gene

expression are reciprocally either neutral or anti-correlated with respect to the active regulatory clusters. Thus, the regulatory element clusters defined by IRF4, XBP1, and BLIMP1 occupancy provide a coherent picture and link in a dichotomous fashion to the key elements of the gene expression network of the ABC to plasmablast transition.

## G9A inhibitor UNC0638 impacts on the efficiency of ASC generation

To explore the potential contribution of coregulators of BLIMP1 to the repression of gene modules during the ABC to plasmablast transition, we focused on EHMT2/G9A, taking advantage of the selective pharmacological inhibitor UNC0638 (46). G9A is an H3K9-directed methyltransferase which can be recruited by BLIMP1 and catalyzes the repressive H3K9me2 modification.

We initially evaluated the impact of UNC0638 treatment on the functional characteristics of the ABC to plasmablast transition. We identified a dose of UNC0638 that was sufficient to impair features of phenotypic differentiation and to impact on global H3K9me2 levels across the 72 h of culture (Fig S10A). G9A inhibitors have been reported to induce autophagy; this was also observed in the B-cell response to this G9A inhibitor with rapid induction of features

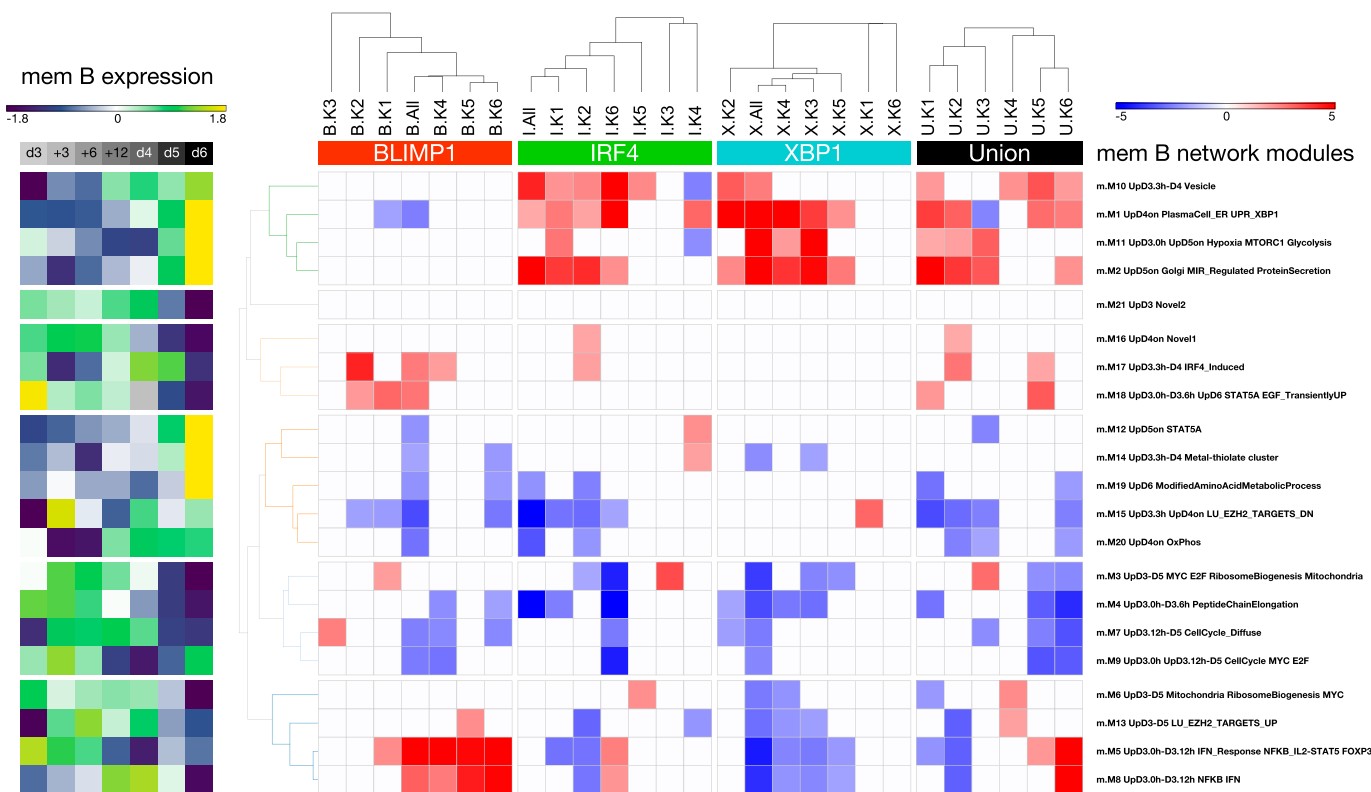

**Figure 6. Integration of gene regulatory modules of the activated B-cell to plasmablast transition with TF occupancy patterns and epigenetic state.**
Signature enrichment heat map displaying the enrichment/depletion of the memory B-cell parsimonious gene correlation network analysis expression modules (Fig 3) for genes associated with the TF peaks in the K-means clusters of epigenetic state (Fig 5). Significance of association between TF occupancy and genes belonging to a network module is shown as a z-score color scale (−5:blue to +5:red) divided according to hierarchical clustering of K-means modules (top) (z-scores with a P-value > 0.05 were set to 0). Results ordered left to right: BLIMP1 K-means clusters (B.All & B.K1-6), IRF4 K-means clusters (I.All & I.K1-6), XBP1 K-means clusters (X.All & X.K1-6), and Union K-means clusters (U.All & U.K1-6). Module identity is indicated to the right. Median expression pattern of the module across the time course illustrated as a z-score (−1.8: dark blue to 1.8: yellow) (left).

consistent with autophagy in ABCs after UNC0638 treatment (Fig S10B and C). Autophagy can have a protective role in PC differentiation (47, 48), and despite the stress response, phenotypic differentiation showed limited differences with modest impairment in the down-regulation of CD20 and the up-regulation of CD38 (Fig S10D). Furthermore, consistent with a delayed impact as would be anticipated with an epigenetic mechanism cell division was modestly impaired at day 4 (24 h of treatment) and delayed by one generation at day 5 and 6 where the population mode for UNC0638-treated cells fell at generation 6 rather than 7 (Fig S10E). Notably, functional differentiation was largely maintained, with day 6 cells seeded at equivalent densities showing equal numbers of ASCs (Fig S10F). Thus, at a dose of G9A inhibitor that reduced global methylation of H3K9me2 and induced the characteristic autophagic stress response, functional ASC differentiation remained largely intact providing a suitable condition in which to assess consequences of acute G9A inhibition.

### A subset of BLIMP1-bound genes linked to the ABC state is responsive to UNC0638

Despite the rapid induction of cellular stress response, there were no significantly differentially expressed genes after UNC0638 treatment until 24 h after treatment, and substantial numbers of differentially expressed genes did not appear until 72 h in day 6 plasmablast (Fig 7A and Table S6). The impact of UNC0638 on autophagic stress response, which occurred early is, therefore, not explained in this system by differential gene regulation. Furthermore, the delayed impact on gene expression is consistent with an epigenetic effect in this rapidly dividing population. We therefore tested BLIMP1 occupancy and associated epigenetic marks at the day 6 time point of maximal difference in gene expression. We aimed to evaluate the H3K9me2 and H3K27me3 repressive marks along with BLIMP1 and active histone marks. However, although BLIMP1 ChIP-seq and active histone marks were informative, data sets for H3K9me2 and H3K27me3 showed a high level of background and were consequently excluded. Consistent with the maintenance of functional ASC differentiation, BLIMP1 occupancy was globally unimpaired in the presence of UNC0638 treatment (Fig 7B). Repeating the K-means clustering of all BLIMP1 peak regions (union of control/inhibitor-treated peaks) illustrated that the overall pattern of association of BLIMP1 with active (high H3K4me3 and H3K27ac) or inactive (low H3K4me3 and H3K27ac) regulatory regions was maintained (Fig 7C). H3K4me3 and H3K27ac signals were not reduced in the presence of inhibitor. Thus, G9A inhibitor treatment resulted in a highly selective impact on differential gene expression without globally perturbing the pattern of BLIMP1 occupancy or active chromatin of the differentiating plasmablast population.

We next mapped the gene expression changes onto the PGCNA network for the ABC to plasmablast transition. This demonstrated a focused effect for both genes differentially up-regulated and down-regulated by UNC0638 (Fig 7D and E). Genes that were expressed at higher levels in control conditions at day 6 were significantly enriched among genes in modules up-regulated at the plasmablast stage m.M1, m.M2, and m.M12 and were most strongly enriched among genes in module m.M1 encompassing genes characteristic of the PC state. By contrast, those genes expressed at higher level in the presence of UNC0638 (down-regulated in standard conditions)

were significantly skewed to modules with opposite kinetics during normal differentiation, which are repressed during plasmablast differentiation m.M5, m.M8, and m.M13 and that link to regulatory element clusters associated with BLIMP1 occupancy (m.M5 and m.M8) and relatively inactive chromatin state. Hence, the impact of G9A inhibition on expression is concordant with the regulatory element clusters and consistent with a focused delay in differentiation-related gene expression.

Although UNC0638-mediated G9A inhibition is not selective for the nature of G9A targeting to chromatin, the pattern of gene expression changes supports a functional link with BLIMP1. Integrating differential expression with occupancy showed a significant skewing toward local BLIMP1 occupancy among genes up-regulated in the presence of UNC0638 (twofold-up: 14/32, 44% BLIMP1 bound versus twofold-down: 3/19, 16% BLIMP1 bound). The genes differentially up-regulated included *IL2RA*, which has been previously defined as a BLIMP1 and G9A target in T-cells (49). In addition, the gene set included characteristic TFs of the ABC state *BATF*, *MYB*, and *RUNX3* as well as typical features of the ABC state *CCL22* and *CCR7* (Fig 7D and F) (50). This coherent impact of G9A inhibition on the differentiation program provides evidence for a selective dependency: repression of components of the ABC program that are also associated with local BLIMP1 occupancy.

### IRF4 shifts regulatory element occupancy pattern and gene network association at the ABC toplasmablast transition

Among the genes which failed to be normally repressed on G9A inhibition during the ABC to plasmablast transition, *BATF* was of particular interest because it provides a partner for IRF4 (19, 20). During murine B-cell activation and differentiation detailed analysis of the modes of IRF4 binding has demonstrated preferential usage of AICE sites at earlier stages of B-cell activation (21). Furthermore, co-occupancy by BATF and IRF4 at AICE sites is a feature of a subset of DLBCL which are arrested at the ABC stage (51). In plasmablast, IRF4 occupancy was associated with enrichment of EICE and ISRE variants but with limited evidence of AICE sites (Fig 4). We therefore assessed IRF4 occupancy in human ABCs to address the extent to which AICE-associated binding could be observed at this stage. Because of limitations in cell number and population expansion at the day 3 time point, these experiments were performed from differentiation of total peripheral blood B-cells. This results in a mixed representation of ABCs derived from naïve and memory B-cell fractions, which differs from the enriched memory B-cell derivation in the day 6 plasmablast data. However, the similarities in gene expression patterns, including the repression of BATF that we observed between differentiation of total and memory B-cell fractions (Figs 1 and 2), suggest that the overall patterns of gene regulation are highly similar. We reasoned, therefore, that the distribution and gene regulatory inputs of key TFs would remain broadly comparable, allowing comparison between these day 3 data and the data derived from day 6 plasmablasts.

IRF4 ChIP-seq of such ABCs at day 3 recovered 18,271 occupied sites which overall showed a preference for intronic and intergenic binding, at a significantly greater fraction of sites than observed for IRF4 occupancy at day 6 (Fig S6F). De novo motif analysis recovered AP1 and both AICE motifs along with EICEs and IRF generic motifs as

**Figure 7. G9A inhibition with UNC0638 produces a focused impact on gene expression during plasmablast differentiation.**
**(A)** Graphical representation of differential gene expression across a range of fold-change thresholds from 1.2+ to 2.0+ across a time course after UNC0638 treatment at D3 of memory B-cell differentiation. Number of differentially expressed genes across each fold-change threshold indicated by the color-coded graphical representation, according to the color code on the right of the figure (y-axis: number of significantly differentially expressed genes, x-axis: time point in hours and days). Left graph: genes up-regulated in the absence of inhibitors, right graph: genes up-regulated in the presence of inhibitors. Data derived from three independent donors. **(B)** The overlap of BLIMP1 occupancy in memory-derived plasmablasts in the absence (blue) or presence (brown) of UNC0638 treatment. **(C)** deepTools heat maps of K-means clusters

the most common and significant. Overall, this pattern differed substantially from that observed for IRF4 at day 6 and supported a substantial contribution for AP1-associated IRF4 binding. Notably, a CTCF/BORIS motif was not identified by de novo motif analysis at sites of day 3 IRF4 occupancy.

The extent of overlap between IRF4 occupancy patterns in the ABC and plasmablast data sets was relatively limited with 2,483 sites occupied by IRF4 at both stages, representing 13.6% of IRF4 occupancy in ABCs and 26% of IRF4 occupancy in plasmablasts (Fig 8A). To directly compare the occurrence of different motifs at these sites, we assessed 12 known motifs encompassing minimal IRF, ISRE, EICE, and AICE variant 1 as well as IRF4 co-factor motifs for BATF, SPIB, PU.1, and CTCF. The analysis of these known motifs underlined the difference in IRF4 binding pattern (Fig 8B). Sites occupied in ABCs were the most significantly enriched for BATF and AICE motifs but showed little enrichment for CTCF motifs. Sites shared between ABCs and plasmablasts favored EICE and ISRE binding modes and also showed little enrichment of CTCF motifs. By contrast, sites uniquely occupied by IRF4 in plasmablasts were significantly enriched for CTCF motifs and showed a preference for minimal IRF, EICE, and ISRE over AICE or BATF motifs.

To further explore this differential association, we turned to ChIP-seq data sets derived from cell lines representing neoplastic B-cells transformed either at the ABC or PC stage, that is, ABC-DLBCL and PC myeloma. In both these diseases, IRF4 plays a critical role (52, 53) but would be expected to show differential motif usage and association with binding partners. Indeed, in ABC-DLBCL cell lines, IRF4 occupancy associated with AICE and EICE motifs, as previously shown (51) but with limited enrichment of CTCF motifs (Fig 8B; OCILY_Comb). By contrast, in myeloma cell lines, IRF4 occupancy was linked to ISRE, EICE, and CTCF motifs (Fig 8B; MM_Comb). Interestingly, given the common deregulation of the NFkB pathway in myeloma (54, 55), which provides an upstream driver of BATF expression (56), somewhat greater enrichment of AICE and BATF motifs was observed in myeloma cell lines than in plasmablasts. Thus, in the neoplastic cell lines, the differential associations of IRF4 occupancy seen in the ABC to plasmablast transition were recapitulated, and IRF4 association with binding sites enriched for CTCF motifs was found as a significantly more prominent feature in the PC myeloma than ABC-DLBCL cell lines tested.

The overall difference in IRF4 occupancy pattern for ABCs during differentiation was also evident when analysed by K-means clustering (Fig 8C). Most IRF4 binding in ABCs clustered with regions lacking active promoter marks in plasmablasts (U2.K5 and U2.K6). This contrasted with sites bound by IRF4 in plasmablasts which associated with active histone marks or with CTCF occupancy but showed little evidence of IRF4 binding in ABCs (U2.K1-K3). Most

regions bound by IRF4 in both ABCs and plasmablast corresponded to sites with relatively weak active histone marks in plasmablasts (U2.K4). When integrated with the expression network modules (Fig 8D), the genomic elements selectively occupied by IRF4 in ABCs (U2.K5 and U2.K6) were associated with modules of genes expressed at the ABC stage. These modules of gene expression were repressed in plasmablasts and linked to BLIMP1 regulatory element clusters depleted of active histone marks in plasmablasts (B.K4-B.K6).

Thus, a further dichotomy of TF association is observed as IRF4 occupancy shifts between different sets of regulatory elements in the ABC to plasmablast transition. This shifting pattern broadly recapitulates that observed during murine B-cell activation (21). The IRF4 regulatory elements are enriched in the vicinity of modules of genes expressed either at the ABC or plasmablast stage and are linked to different underlying DNA-binding motif usage. In conjunction with the shift in IRF4 occupancy pattern, and potentially reinforced through repression of *BATF*, BLIMP1 binding becomes enriched in the vicinity of genes expressed at the ABC stage as the genes are repressed, whereas XBP1 occupancy is established at promoters of secretory pathway genes (Fig S11).

## Discussion

In the differentiation of B-cells to the PC state, two primary transitional states are the ABC and the plasmablast. Both of these are ephemeral cell states, the former linked to the extracellular cues driving the activation process and the latter representing the penultimate stage poised for decision between cell death or entry into cell cycle quiescence and the completion of the PC differentiation program. This transition encompasses the cusp between the lymphoid and the antibody-secreting states of the B-cell life cycle. Given the prevailing model of a broadly epistatic relationship between IRF4, BLIMP1, and XBP1 during the establishment of PC differentiation (1), we sought to explore how the reorganizing gene expression network in human B-cells related to the pattern of TF occupancy at the plasmablast state. Our data support the conclusion that these three TFs have quite distinct associations with both epigenetic state and gene regulation.

We have recently established an expression networking tool, PGCNA, which can be effectively applied to time course data sets (5, 41). Here, this approach allows the detailed definition of modules of gene co-expression accompanying the differentiation of B-cells between the ABC and plasmablast states. This illustrates that at the ABC stage, a combination of growth and cell division related programs are dominated by gene signatures associated with MYC

---

derived from the union of BLIMP1 binding sites for standard and UNC0638 conditions and considering associated epigenetic marks as indicated for H3K4me3, and H3K27ac. **(D)** Heat map of genes up-regulated upon G9A inhibition (fold change > 1.8, FDR < 0.05) showing patterns of gene expression as z-scores (−1.8: dark blue to 1.8: yellow) across the differentiation in the absence or presence of UNC0638 treatment. To the right are the genes identified as BLIMP1 bound highlighted with red bars. **(A, E)** Dumbbell graph of the relative enrichment or depletion of differentially expressed genes shown in (A) against the modules of gene expression derived from the memory B-cell parsimonious gene correlation network analysis network in Fig 3. Y-axis shows the order of modules ranked from most significantly enriched for genes up-regulated in the presence of UNC0638 through most significantly enriched in the standard conditions. For each module, the enrichments or depletion are shown for the genes up-regulated in the presence of UNC0638 (brown circles) and genes up-regulated in standard conditions (blue circles). These are plotted against the x-axis displaying Z-score of enrichment/depletion with the vertical dotted red lines indicating the point of FDR corrected significance (*P*-value < 0.05). **(D, F)** Representative tracks for BLIMP1 ChIP-seq in standard and UNC0638-treated samples, as indicated for representative genes selected from (D) whose expression is increased in the presence of UNC0638.

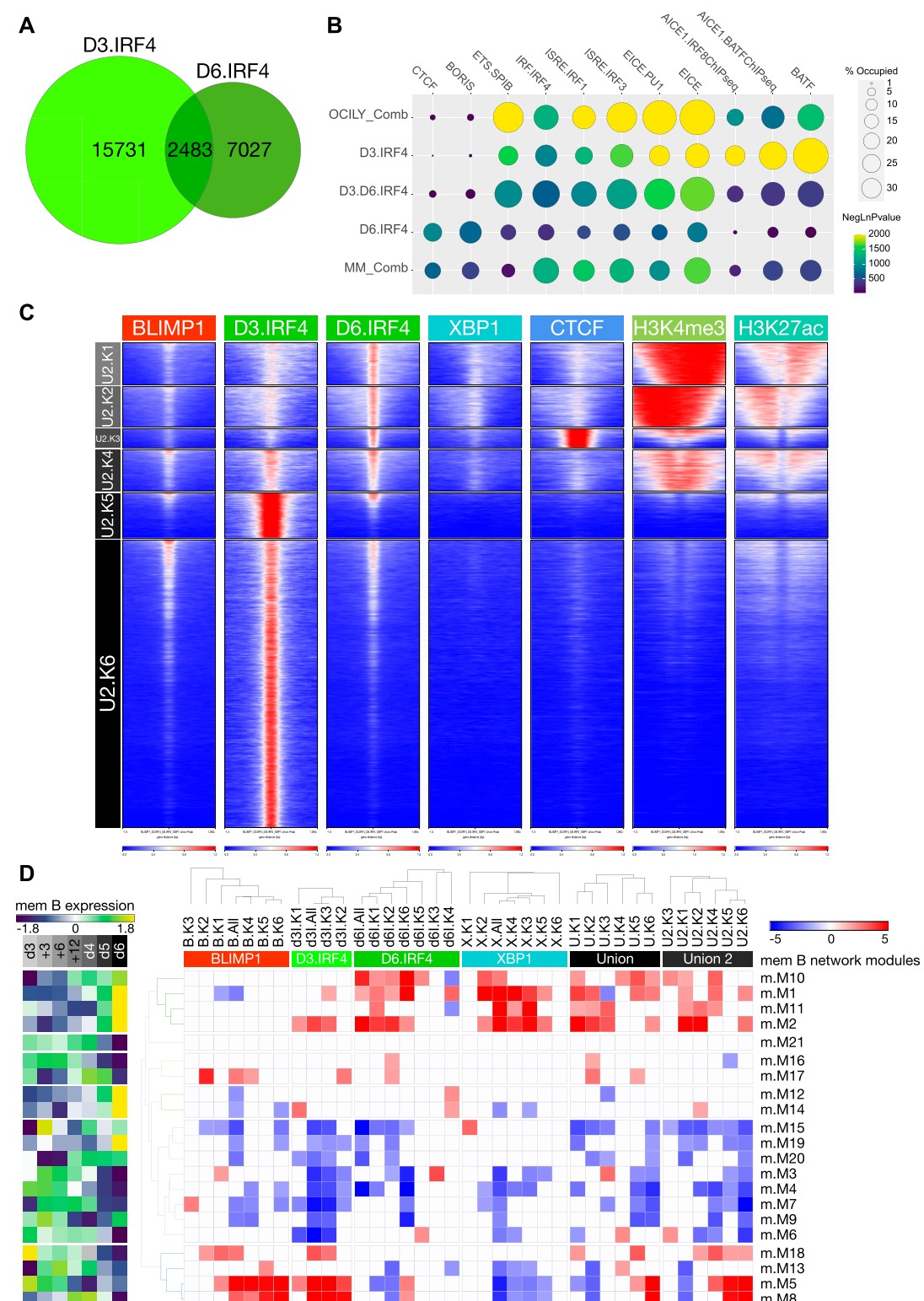

**Figure 8. Differential IRF4 occupancy and gene network associations between the activated B-cell (ABC) and plasmablast states.**
**(A)** Venn diagram of overlap in IRF4 occupancy between ABCs (D3.IRF4 light green) and plasmablasts (D6.IRF4 darker green). **(A, B)** Comparison of known motif enrichments at sites bound by IRF4 as shown in part (A) either in ABCs only (D3.IRF4), ABCs and plasmablasts (D3.D6.IRF4), or plasmablasts only (D6.IRF4) and compared with IRF4 occupancy in cell lines representative of transformation at the ABC state (OCILY_Comb derived from OCI-LY3 and OCI-LY10 ABC-DLBCL lines) and plasma cell myeloma (MM_Comb derived from H929 and U266 malignant myeloma cell lines). Enrichment of known motifs indicated across the top is illustrated as percentage of sites with motif match (circle diameter–top right) and heat map color code (dark blue to yellow, ceiling set at –logP ≥ 2,000, bottom right). **(C)** deepTools heat maps of

and E2F along with sustained expression of input signaling in the form of immediate early genes as well as NFκB and STAT pathway target genes. The dependence of these modules of gene expression on the input signal is suggested by their rapid loss in expression within a few hours of removal from the activating conditions. This contrasts with the dynamics of MYC- and E2F-related modules, which after a brief dip in expression recover to sustain expression during the ensuing proliferative phase. Although we cannot conclude from the time course data that the sustained input signals are in themselves acting to delay the transition into the proliferation and differentiation phase of the ABC to plasmablast transition, the sequence of gene expression changes would be consistent with such a model. This would potentially be explained by the recent demonstration that sustained REL activation can delay PC differentiation (57). It is conceivable that a balance in favor of cell growth over division and differentiation would persist if the input signaling from CD40 acting via REL was maintained and that the principle trigger for transition into the ABC to plasmablast phase, in which commitment to differentiation becomes progressively more pronounced, is the removal of this input signal. Our results would also be consistent with recent murine in vivo studies demonstrating that MYC-dependent growth signals delivered in response to T-dependent help can establish a division potential before release into a phase of rapid cell division (58, 59). At the ABC state, therefore, the B-cells are primed for rapid division and division-linked differentiation.

During the subsequent transition toward the secretory state, integrating modular patterns of gene expression with global TF occupancy and epigenetic patterns supports a general segregation between IRF4 and XBP1 as activators of gene expression and BLIMP1 as a repressor during this transitional window. The key modules that identify the PC state are significantly enriched for association with local occupancy by IRF4 and XBP1 in association with active chromatin at the plasmablast stage. BLIMP1 occupancy is significantly associated with genes that are repressed during the ABC to plasmablast transition and overall BLIMP1 occupancy is anti-correlated with the key module characteristic of the plasmablast/PC state (e.g., m.M1) or modules encompassing the wider secretory pathway in human plasmablasts. Thus, we failed to find strong evidence in support of an expanded role for BLIMP1 in the direct control of secretory pathway components, as has been shown in murine PCs (13, 26). The reason for this difference is uncertain but may lie in the fact that we have focused on the transitional plasmablast differentiation stage rather than on a quiescent PC population. It is possible that during the initial establishment of differentiation, the repressive function of BLIMP1 predominates, whereas an expanded role for BLIMP1 in controlling secretory pathway genes may become evident at later stages of human PC differentiation, as further secretory optimization accompanies cell cycle exit (5). Furthermore, technical differences in chromatin immunoprecipitation (ChIP)

procedure as well as differences in stringency of peak calling could contribute to disparities in representation of lower affinity interactions. We note that at an individual gene-level BLIMP1 does occupy active regulatory elements of secretory pathway genes in human plasmablasts. For example, of 45 secretory pathway genes linked to BLIMP1 regulation in murine PCs (13), seven (SIL1, DPAGT1, ERP44, HSP90B1, PDIA3, PPP1R15A, and SLC33A1) showed evidence of BLIMP1 binding in human plasmablasts at the level of peak calling stringency used in our analysis (identifying 4323 BLIMP1 peaks). However, in this context, of the more discrete set of XBP1 targets identified (605 peaks), 24/45 of these secretory pathway components were occupied by XBP1. Indeed, although XBP1 ChIP-seq returned only a limited set of target genes at the plasmablast stage, these are highly correlated with the secretory gene expression modules in these cells.

Our understanding of the importance of XBP1 to the establishment of the PC fate has shifted from central integration of the ER stress response with the differentiation process via XBP1 (60, 61), toward a more specialized role for XBP1 in optimization of PC secretory functions (14, 16, 62). Our data demonstrate that in human plasmablasts, XBP1 selectively binds to active promoters of many secretory pathway genes. The kinetics of expression of these genes is in parallel with, rather than subsequent to, that of other characteristic phenotypic components of the PC state. Moreover, the primary module (m.M1_PlasmaCell) of gene expression with which XBP1 occupancy associates, initiates expression from 24 h after release from CD40 signal, and is highly enriched for genes belonging to the UPR and ER-stress response. Although an additional wave of UPR/ER-stress responsive gene regulation occurs as human plasmablasts complete the differentiation to the quiescent PC state (5), the data presented here using a spliced XBP1 specific antibody (43) point to a significant contribution for XBP1 activation at the initial point of secretory pathway commitment. Murine models of conditional XBP1 deletion have demonstrated that a contribution from XBP1 is not an absolute pre-requisite for the generation of phenotypic PCs but is required for establishment of optimal function (14, 16, 62). These results might be explained by redundancy because additional TFs linked to the ER stress response are regulated during PC differentiation such as ATF6 and CREB3L2 (63, 64).

At the plasmablast stage, the strongest associations for BLIMP1 were with occupancy at regulatory elements lacking active chromatin marks and in the vicinity of genes expressed at the ABC stage that are repressed in plasmablasts. Repression of a subset of these genes is sensitive to inhibition of the histone methyltransferase G9A, supporting a specific role for this epigenetic modifier in repression of the ABC state. Among genes sensitive to G9A inhibition was the transcriptional regulator BATF, which provides a partner of IRF4 in a range of cell lineages at a distinct set of regulatory elements (19, 20). Expression of BATF is a key features of the ABC state

K-means clusters derived from D3.IRF4-bound regions alongside D6.IRF4 and sites bound by BLIMP1, CTCF, H3K27ac, and H3K4me3 in day 6 plasmablasts. **(C, D)** Integration of gene regulatory modules of the ABC to plasmablast transition with TF occupancy patterns and epigenetic state as in Fig 6 but with the added inclusion of K-means clusters for D3.IRF4-occupied regions alone, and the union of all occupied regions including D3.IRF4 (Union 2) as in (C). Heat map displays the enrichment/depletion of TF peaks in the K-means clusters relative to the memory B-cell parsimonious gene correlation network analysis expression modules. Significance of TF occupancy versus genes belonging to a network module is shown as a z-score color scale (−5: blue to +5: red) divided according to the hierarchical clustering of K-means modules (top) (z-scores with a P-value > 0.05 were set to 0). Module identity is indicated on the right, and median expression pattern of the module is shown across the time course as a z-score on the left (−1.8: dark blue to 1.8: yellow).

in B-cell lymphomas in which it provides an important partner for IRF4 ([50], [51]). As a direct target of NFkB signals ([56]), BATF is implicated as one of the transcriptional determinants of B-cell fate in response to CD40 ligation ([65]). A shift in IRF4 occupancy from a BATF-associated AICE motif pattern to an ISRE- and EICE-dominated mode of DNA binding has been identified as a key transition in murine B-cell activation and PC differentiation ([21]). We were therefore interested to determine whether this was also a feature of the ABC to plasmablast transition. Indeed, this was the case with a clear shift of IRF4 binding away from an AP1 dominant pattern between the two cell states. Integration with the gene expression network exemplified how the change in IRF4 binding correlated with a shift from occupancy in vicinity of genes expressed in ABCs to genes expressed in plasmablasts.

A caveat to this analysis is that the data sets used in this comparison derive on the one hand from differentiation of enriched memory B-cells at the plasmablast stage, and on the other hand from total peripheral blood B-cells for ABCs at day 3. This arose because of limitations in cell population expansion at the day 3 relative to day 6 stage of the differentiation system. Because we observed very similar patterns of overall gene regulation between total and memory B-cell enriched differentiations at the level of gene expression, we reasoned that the comparison remained meaningful at the level of a population level shift in IRF4 occupancy pattern during B-cell differentiation. This conclusion is further supported by the similarities observed in ABC-DLBCL cell line data. Our understanding of heterogeneity in human peripheral blood B-cell populations is rapidly increasing ([66]). It will be interesting in future to address whether particular B-cell subsets show selective differences in gene regulation and key TF occupancy during the ABC to plasmablast transition.

Several factors may contribute to *BATF* repression during the ABC to plasmablast transition; this includes loss of CD40 signal–mediated NFkB activation, along with potential transcriptional repression by BLIMP1 in association with G9A. Although we have not performed detailed mechanistic studies to further substantiate this linkage, it is plausible that BLIMP1 expression contributes to the repression of *BATF*, thus altering the nature of available IRF4 TF partners. This mirrors the regulatory arrangement previously identified for the alternate IRF4 partner SPIB ([22]). At the ABC stage, SPIB can provide a significant DNA-binding partner for IRF4 at EICE motifs. Repression of both *BATF* and *SPIB* by BLIMP1 would reinforce a shift in IRF4 binding, initiated by loss of the CD40 input signal in this model. IRF4 occupancy at EICE motifs could be selectively maintained through differential control of PU.1 (encoded by *SPI1*), which is the other ETS-factor partner for IRF4 at EICEs and is not a direct target of BLIMP1. As IRF4 binding shifts away from its AICE-associated pattern, BLIMP1 binding is established in association with modules of genes previously under the influence of IRF4 at the ABC stage. The combination of regulation of IRF4-binding partners and change in direct input would provide the potential for reinforced repression of the associated genetic programs by BLIMP1 and recapitulates elements of gene regulatory patterns seen in other differentiation decisions.

A notable finding in relation to IRF4 occupancy in human plasmablasts is a significant association with CTCF binding, which is substantially different in proportion for that observed for either BLIMP1 or XBP1. Indeed, co-occupancy by IRF4 and CTCF is linked to

motifs that disfavor BLIMP1 binding. Although a low level of enrichment of CTCF motifs can be seen at IRF4-bound regions in ABCs and ABC-DLBCL, this is significantly lower than in plasmablasts or myeloma cell lines, suggesting that the association is linked to differentiation. IRF4 expression levels in T-cells correlate with signal intensity and cell fate choice, with high expression linked to effector cell fate ([67]). An association between IRF4 and CTCF has been identified in Th17 cells at sites co-occupied with STAT3, BATF, and BRD2 ([68]). Of note in plasmablasts, at CTCF/IRF4 co-occupied sites, EICE motifs predominate which suggests a divergence from the binding mode in the Th17 context. It will be intriguing to explore whether the link between IRF4 and CTCF reflects a particular contribution to long-range chromatin interactions in the ASC effector state.

In summary, our integrated analysis illustrates the connection between reorganizing gene expression and TF binding at the ABC to plasmablast transition in human B-cells and reinforces both integration and functional segregation between IRF4, BLIMP1, and XBP1 during this process.

# Materials and Methods

### Reagents

For the in vitro cell stimulation and maintenance, the following reagents were used: human IL-2 (Roche); IL-21 (PeproTech); goat antihuman F(ab')$_2$ fragments (anti-IgM & IgG) (Jackson Immuno-Research); lipid mixture 1 chemically defined (200×) and MEM Amino Acids Solution (50×) (Sigma-Aldrich); for G9A inhibition, UNC0638 (Cayman Chemical); and for cell proliferation: CFSE (Sigma-Aldrich).

### Donors and cell isolation

Peripheral blood was obtained from healthy donors after informed consent. Mononuclear cells were isolated by Lymphoprep (Axis Shield) density-gradient centrifugation. Total B-cells were isolated by negative selection with the Memory B-cell Isolation Kit (Miltenyi Biotec). Memory-enriched B-cell fractions were isolated by negative selection after incubation of total, negatively selected B-cell fractions with CD23 Biotin and anti-Biotin Microbeads (Miltenyi Biotec).

### Cell cultures

24-well flat-bottom culture plates (Corning) and IMDM supplemented with GlutaMAX and 10% heat-inactivated FBS (HIFBS; Invitrogen) were used. Day 0 to day 3: B-cells were cultured at 2.5 × 10$^5$/ml with IL-2 (20 U/ml), IL-21 (50 ng/ml), and F(ab')$_2$ goat antihuman IgM & IgG (10 μg/ml) on γ-irradiated CD40L expressing L cells (6.25 × 10$^4$/well). Day 3 to day 6: at day 3, the cells were detached from the CD40L L-cell layer and reseeded at 1 × 10$^5$/ml in media supplemented with IL-2 (20 U/ml), IL-21 (50 ng/ml), Hybridomax hybridoma growth supplement (11 μl/ml), lipid mixture 1, chemically defined and MEM Amino Acids Solution (both at 1× final concentration). For UNC0638 experiments, ABCs were treated

at day 3 with inhibitor at the indicated concentration (generally 2 µM), vehicle control (DMSO), or with standard conditions as indicated. The cells were sampled at the indicated time points without further addition of inhibitor. NCI-H929 and U266 cells (DSMZ) were cultured in RPMI1640 media and OCI-LY3 and OCI-LY10 (Prof R. E. Davis lab) in IMDM with GlutaMAX (Life Technologies), each containing 10% heat inactivated fetal calf serum (51). Cell lines identity was confirmed using short tandem repeat profiling.

### Flow cytometric analysis and microscopy

Cells were analysed using four- to six-color direct immunofluorescence staining on a BD LSR II flow cytometer (BD Biosciences). Antibodies used were as follows: CD19 PE (LT19) and CD138 APC (B-B4; Miltenyi Biotec); CD23 APC (M-L233), CD27 FITC (M-T271), CD38 PE-Cy7 (HB7; BD Bioscience); and CD20 efluor V450 (2H7; eBioscience). Controls were isotype-matched mouse mAbs. Dead cells were excluded by staining with 7-AAD (BD Biosciences). Autophagy was detected with the Cyto-IDAutophagy Detection Kit (Enzo Life Sciences). Absolute cell counts were performed with CountBright beads (Invitrogen). Cell populations were gated on forward scatter (FSC) and SSC profiles for viable cells determined independently in preliminary and parallel experiments. Analysis was performed with BD FACSDiva Software 8.0 (BD Biosciences) and FlowJo v10 (FlowJo LLC).

### Gene expression analysis

RNA was extracted with TRIzol (Invitrogen) and amplified using Illumina TotalPrepTM-96 RNA Amplification Kit (Life Technologies). Resultant complementary RNAs were hybridized onto HumanHT-12 v4 Expression BeadChips (Illumina) according to the manufacturer's instructions, scanned with the Illumina BeadScanner, and initial data processing carried out using the Illumina GenomeStudio. Expression data were derived from samples of three independent donors. For details of normalization and analysis, please see Supplemental Data 1.

### PGCNA and signature enrichment analysis

See Supplemental Data 1, for details of PGCNA; in brief, informative genes are used to calculate Spearman's rank correlations for all gene pairs. For each gene (row) in a correlation matrix, only the three most correlated edges per gene are retained. The resulting matrix M is made symmetrical. The correlation matrices are clustered using a community detection algorithm and the best (judged by modularity score) used for downstream analysis (41). Gene signature analysis for modules was performed using a hypergeometric test against a curated set of signatures (69, 70, 71, 72, 73).

### Western blot, ChIP, and ChIP-seq

At the indicated time points, primary cells were harvested, washed in PBS, and lysed in Laemmli buffer to generate whole cell lysates. Western blots were performed using the following antibodies: BLIMP1 (R23) (74), H3 (ab1791), H3K9me2 (ab1220), H3K27me3 (07-449), and autophagy detection kit (Cell Signaling). ChIP was performed as described (42). Antibodies used were BLIMP1 (R21) (25, 74), IRF4 (sc-28696X), XBP1 (619502; BioLegend), CTCF (07-729), H3K4m3 (04-

745), H3K9me2 (ab1220), H3K27me3 (07-449), and H3K27Ac (ab4729). ChIP-seq libraries were prepared using the MicroPlex Library Preparation Kit (Diagenode) or NEBNext ChIP-seq for IRF4 day 3 samples, size-selected using AMPure XP beads (Beckman Coulter), and run on an Illumina Hiseq 2500 or NextSeq for day 3 IRF4 samples. ChIP-seq data for the day 6 plasmablasts are derived from individual samples derived from memory B-cells, except for XBP1 at day 6 and IRF4 at day 3 which were derived from differentiation of total peripheral blood B-cells. Cell line data are derived from duplicate samples each of the lymphoma lines OCI-LY3 and LY10 and single samples of myeloma lines H929 and U266. Cell line data for lymphoma and myeloma samples were merged independently during analysis to define shared peaks in each cell line context. IRF4 ChIP-seq for day 3 ABCs was performed from two independent samples with data merged during analysis to define shared peaks.

### ChIP-seq data analysis and motif detection

For more detail, see Supplemental Data 1. Trimmed reads were aligned with Bowtie2 (75) and analysed for peaks using GEM and MACS2 with overlapping peak sets retained (76, 77). Peak overlaps were determined using a clustering approach such that any peak centre <250 bp from an index peak centre were considered part of an overlapping cluster (See Supplemental Data 1). De novo motif detection was performed with HOMER (78).

The high-confidence peak sets for BLIMP1, IRF4, and XBP1 along with the Union set (overlap of individual high-confidence BLIMP1, IRF4, and XBP1 peaks) were analysed. Peaks were normalised and extended to their estimated fragment length. Scores per region were calculated for a ±1,000-bp region. The resulting matrix was K-means–clustered and visualised, see Supplemental Data 1 for details.

### Ethical approval

Approval for this study was provided by the UK National Research Ethics Service via the Leeds East Research Ethics Committee, approval reference: 07/Q1206/47.

## Data Availability

Data sets are available with GEO accession GSE142492 (gene expression) and GSE142493 (ChIP-seq). Interactive visualisations and additional analyses are available at https://mcare.link/abctopb.

## Supplementary Information

## Acknowledgements

This work was supported by Cancer Research UK program grant C7845/A17723 & C7845/A29212, Cancer Research accelerator award C355/A26819, Blood Cancer Research UK project grant 14034, and a studentship of the Sultanate

of Oman (to M Al-Maskari). We thank Sophie Stephenson for assistance with flow cytometry analysis and Ulf Klein for critical review of the manuscript.

## Author Contributions

M Cocco: conceptualization, formal analysis, investigation, and methodology.

MA Care: conceptualization, resources, data curation, software, formal analysis, visualization, methodology, and writing—review and editing.

A Saadi: conceptualization, formal analysis, funding acquisition, validation, investigation, visualization, project administration, and writing—original draft, review, and editing.

M Al-Maskari: formal analysis and investigation.

G Doody: conceptualization, funding acquisition, investigation, and writing—review and editing.

R Tooze: conceptualization, formal analysis, funding acquisition, validation, investigation, visualization, project administration, and writing—original draft, review, and editing.

## Conflict of Interest Statement

The authors declare that they have no conflict of interest.

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
