## [Reviewer comments · Life Science Alliance]

Life Science Alliance

A dichotomy of gene regulatory associations during the activated B-cell to plasmablast transition

Mario Cocco, Matthew Care, Amel Saadi, Muna Al-Maskari, Gina Doody, and Reuben Tooze
DOI: <https://doi.org/10.26508/lsa.202000654>

Corresponding author(s): Reuben Tooze, University of Leeds

Review Timeline:	Submission Date:	2020-01-22
	Editorial Decision:	2020-02-26
	Revision Received:	2020-07-10
	Editorial Decision:	2020-07-28
	Revision Received:	2020-08-12
	Accepted:	2020-08-12

Scientific Editor: Shachi Bhatt

Transaction Report:

February 26, 2020

Re: Life Science Alliance manuscript #LSA-2020-00654-T

Prof. Reuben M Tooze
University of Leeds
Leeds Institute of Medical Research
St James's University Hospital
Beckett Street
Leeds LS9 7TF
United Kingdom

Dear Dr. Tooze,

Thank you for submitting your manuscript entitled "A dichotomy of gene regulatory associations during the activated B-cell to plasmablast transition" to Life Science Alliance. The manuscript was assessed by expert reviewers, whose comments are appended to this letter.

As you will see, the reviewers appreciate the dataset provided, and they provide constructive input on how to further strengthen your work and, importantly, ensure robustness. We would thus like to invite you to submit a revised version of your manuscript to us, addressing the individual points raised. The issues raised by reviewer #2 regarding data representation and uncertainty of robustness of the analyses need to get addressed in a really good way, and we would be happy to discuss the individual revision points further with you should this be helpful.

Thank you for this interesting contribution to Life Science Alliance. We are looking forward to receiving your revised manuscript.

Sincerely,

B. MANUSCRIPT ORGANIZATION AND FORMATTING:

Reviewer #1 (Comments to the Authors (Required)):

The manuscript of Cocco and colleagues investigates the gene regulatory networks underpinning the transition of human activated B cells to antibody secreting plasmablasts. They perform a highly granular time course of differentiation in cultured B cells, commencing with either naïve or memory

B cells. They also perform extensive ChIPseq analysis on the 3 transcription factors, XBP1, BLIMP1 and IRF4 known to be involved in the differentiation and function of plasmablasts. Matched data for histone modifications and CTCF is also provided. Integration of the gene expression modules with the ChIPseq data, shows that IRF4 and XBP1 predominantly link to genes induced in plasmablasts and BLIMP1 associates with repressed B cell genes.

Overall this is a valuable dataset for the study of human humoral immunity, a subject of high clinical relevance. Although much of the data and conclusions parallels that found in similar studies in the murine system (e.g. Ref17), some potential species specific differences have arisen, such as in BLIMP1 function. A limitation of the study, it that beyond a modest association of Blimp1 binding and G9A activity there isn't any attempt to test predictions from the dataset.

I have the following specific comments that should be addressed.

1. Although BLIMP1 and XBP1s are plasmablast specific, IRF4 is expressed and functional in activated B cells. As the authors have not separated Ab secreting plasmablasts from activated B cells at the day 6 timepoint, some of the IRF4 signal may come from B cells. Have the authors performed IRF4 ChIPseq on day 3 activated B cells. On a similar front it is not clear from the data in fig S4 and 11 what proportion of day 6 cells are Ab secreting.
2. The Roy et al 2029 Immunity paper on NFkB gene usage, is highly relevant to this study and should be referenced and discussed.
3. This study is conducted entirely in culture (akin to Roy et al and Ref 17). For example, no transcriptomic datasets from in vivo plasma cells were examined. The authors should consider this variable in interpreting any differences between mouse and human functions on BLIMP1 and XBP1, as in the mouse cells the consequences of the loss of either factor on the UPR was strongest in mature long-lived plasma cells.

Reviewer #2 (Comments to the Authors (Required)):

In this manuscript Cocco et al. seek to fine map the gene expression changes, transcription factors, and epigenetic processes that occur during human B cell to plasma cell differentiation. The authors focus on two key phases when B cells first transition into an activated B cell state and ultimately from there into plasmablasts that secrete antibodies. Using gene expression profiling and PGCNA network analysis tools that this group has developed sets of coregulated gene modules were defined that match many of the functions of cells at each phase that have been defined in the literature in murine systems. Contrasting the dynamics of this process in naïve and memory B cells provides a nice comparison to show the similarity of programming that all B cells undergo during terminal differentiation. Further analysis of three key transcription factors IRF4, XBP1, and Blimp-1, and epigenetic modifications by ChIP-seq helped define regulatory modules. New concepts for human B cells include the notion of a shared IRF4/XBP1 regulatory module, that Blimp-1 primarily acts as a repressor, and the colocalization of IRF4 and CTCF motifs. To further define the role of epigenetic repression a G9A inhibitor was used to deplete the H3K9me2 histone modification during the later stages of differentiation and the ChIP-seq analysis repeated. While loss of H3K9me2 did not affect Blimp-1 binding, it did affect the ability to extinguish the activated B cell program transcriptional modules, although the phenotypic consequences on antibody secretion were not clear.

This descriptive study provides a wealth of new data for human B cell differentiation that suggests

some intriguing differences from the murine systems. However, some concerns, including the ChIP-seq data quality, need to be addressed.

Major Concerns

1. What is mainly presented in the figures is a global view of differences between stats based on the modules identified by PGCNA. Many of the figures are low resolution and are hard to read labeling even when expanded on a computer. For example, none of the heatmap scales can be read and the time point annotation above them are equally blurry. Additionally, example genes and example gene strips for ChIP-seq data need to be included in the primary figures to provide the reader examples genes to understand the patterns. Without boiling complex patterns and concepts down to an example gene or loci readers will be lost. Consider moving Supplemental Fig 3 into Fig 1 and replacing one or more of the panels if necessary.
2. No data is presented that allow the evaluation of the ChIP-seq data quality. The inclusion of gene strips for loci to show transcription factor binding sites that fit each module and the histone patterns would help to evaluate the specificity of enrichment. One concern is the enrichment in Fig 4 of the repressive histone marks H3K27me3 and H3K9me2 in all modules along with the other activating marks. To me this suggests these ChIPs did not work and are not specific. Modules U.k1-4 all have both activating and repressive marks and this needs to be reconciled to be classified as active gene clusters.
3. Were any ChIP controls used during the ChIP for H3K9me2 in samples that were treated with the G9a inhibitor? The H3K9me2 data in Fig 6 does not look specific with high background and it is known that in conditions with reduced antibody targets that background enrichment can be increased. The use of spike-in standards that several companies sell is a good workaround for some of these antibody issues. As mentioned above, Fig 6C shows enrichment of H3K9me2 at regions that also have activating marks. This is a major concern along with the potential low signal to noise observed in the gene strips.
4. The primary methods (not supplement) should state how many biological replicates were used for all experiments. For example, there are no biological replicates for the ChIP-seq data. While going back to repeat every ChIP is not necessary this should be clearly stated. Additionally, the potential that false-negative results in the data, potentially from low ChIP quality as mentioned above, could explain some of the failure to identify some of the genes regulated by Blimp-1 in mouse plasmablasts.

Minor Concerns:

1. Consider labeling motifs in Fig 3C to easily let the readers know which match the known IRF4 composite motifs (AICE, EICE, ISRE).
2. Improve the overall clarity of figures so a high-resolution supplemental figure is needed to see fine details of specific modules.

Point by point rebuttal LSA manuscript LSA-2020-00654-TR "A dichotomy of gene regulatory associations during the activated B-cell to plasmablast transition".

Reviewer #1 (Comments to the Authors (Required)):

We thank the reviewer for their assessment of the manuscript as providing a "*valuable dataset for the study of human humoral immunity, a subject of high clinical relevance.*"

We agree that the conclusions parallel that found in similar studies in the murine system, but we argue that the integrative analysis of all three of the central transcriptional regulators of the plasma cell state with detailed gene expression time courses provides significant novel insight. Specifically, that this approach in a data-led fashion supports the dichotomy between activatory and repressive inputs at the plasmablast stage for IRF4/XBP1 and BLIMP1, respectively.

1.1 "*Although BLIMP1 and XBP1s are plasmablast specific, IRF4 is expressed and functional in activated B cells. As the authors have not separated Ab secreting plasmablasts from activated B cells at the day 6 timepoint, some of the IRF4 signal may come from B cells.*"

We agree that the approach we have taken based on analysis at specific time points rather than based on sorting of phenotypic subsets, generates a population level assessment with potential contribution from minor fractions with retained B-cell features.

We have taken this approach to allow direct integration with the expression data at the end of the time course. This would not be equivalent if performed on the one hand from total differentiating populations and on the other hand from sorted cells and would intrinsically bias the association toward the modules of genes expressed in the phenotypic subset. Since at the expression level we aim to follow the whole population over time, cell sorting would be challenging to do particularly at early time points. Thus in our opinion the integration with the sequential gene expression time course necessitates derivation of ChIP-seq at an equivalent population level in this study.

At a gene expression level the retention of the B-cell state is low at day-6. This is coupled with the fact that phenotypically plasmablasts represent 80% of the population. While cells with relative retention of the B-cell state will be contributing to the overall IRF4 occupancy pattern, this is nonetheless very different at day-6 from that observed in activated B-cells as demonstrated conclusively by the new data we have included to address the reviewers next point (1.2).

1.2 "*Have the authors performed IRF4 ChIPseq on day 3 activated B cells.*"

We had performed this analysis and have now included these data as suggested by the reviewer. As discussed above this supports the overall conclusions of the manuscript in its revised form. The data are presented in new Figure 8 and in Supplemental Figure 6, and in the text line 534 on p26 to line 604 on p29. The analysis clearly demonstrates the shift in IRF4 occupancy in line with the model of Sciammas et al in murine ASC differentiation. IRF4 has an AP1/AICE dominated pattern in ABCs at day 3 and shifts away from this in the day 6 population. Indeed, this correlates well with loss of BATF expression and identification of BATF as a potential BLIMP1 target which is sensitive to G9A inhibition (Figure 7). Furthermore, the data also demonstrate the shift to a CTCF-associated pattern in plasmablasts. Additional discussion is included on p34/35.

We have further supported these conclusions by including data and analysis of IRF4 ChIP-seq from DLBCL cell lines OCI-LY3 and OCI-LY10 as well as myeloma cell lines NCI-H929 and U266 (Figure 8B). These data illustrate how the separation in binding pattern observed between ABC and PB state in B-cell differentiation including differences in CTCF association is also maintained in neoplastically transformed cell lines related to these stages.

We believe that the analyses in new Figure 8 substantially extend the overall conclusions of the manuscript and support the shifting input delivered by IRF4, which in a closely related setting was also observed in the murine data of the Sciammas lab. We note however that the latter analysis did not include a comparison to BLIMP1, XBP1 and CTCF and therefore while undoubtedly fully supportive of the Sciammas model, our data we would argue also add to this by illustrating the differential relationships of the other factors.

1.3 *“On a similar front it is not clear from the data in fig S4 and 11 what proportion of day 6 cells are Ab secreting.”*

We have sought to directly address this point with data we had available (duplicate samples) and included in new Supplemental Figure 3. In ELISpot assays seeded with 2000 cells we observed 900 spots from IgM and IgG combined. We have not assayed IgA in this context. We also do not have an internal control for the fraction of the 2000 cells that functionally survives the seeding process, but we conservatively estimate that close to 50% of the cells are functionally ASCs at day-6 within the limitations of these assays. While by flow cytometry 80% of the cells have adopted a CD38^{hi} phenotype. These changes appear on p16 line 305-314 in the text.

2. *“The Roy et al 2029 Immunity paper on NFkB gene usage, is highly relevant to this study and should be referenced and discussed.”*

We agree entirely and apologise for the omission, it was in fact included in an earlier draft and this is now referenced and discussed. These changes appear in p31 line 634-639 of the discussion.

3. *“This study is conducted entirely in culture (akin to Roy et al and Ref 17). For example, no transcriptomic datasets from in vivo plasma cells were examined. The authors should consider this variable in interpreting any differences between mouse and human functions on BLIMP1 and XBP1, as in the mouse cells the consequences of the loss of either factor on the UPR was strongest in mature long-lived plasma cells.”*

We accept that we have not analysed transcriptomic data from in vivo plasma cells. We have in context of the development of the model system done so previously and have shown that there are high degrees of similarity in the model to ex vivo human bone marrow plasma cells. But, we fully accept this point. We have substantially rewritten the conclusion to reflect the inclusion of additional data and this point in particular. We mention the differences from the murine studies as these are of interest but point out that there are significant technical differences in particular our focus here on a transitional plasmablast state rather than quiescent plasma cells. These changes appear in p32 line 654-663 of the discussion.

Reviewer #2 (Comments to the Authors (Required)):

We thank the reviewer for the summary that the manuscript presents *“new concepts for human B cells include the notion of a shared IRF4/XBP1 regulatory module, that Blimp-1 primarily acts as a repressor, and the colocalization of IRF4 and CTCF motifs.”* And that the *“study provides a wealth of new data for human B cell differentiation that suggests some intriguing differences from the murine systems.”* As noted above we have addressed all the specific points made.

Major Concerns

1.1 *“Many of the figures are low resolution and are hard to read labeling even when expanded on a computer. For example, none of the heatmap scales can be read and the time point annotation above them are equally blurry.”*

We have done as best we can to address this point within the limitations of figure formats. We have uploaded high quality figures without blurring and have increased fonts where possible. We agree it is difficult to generate figures which both provide a summary of the complex data, are navigable and can be resolved to fine detail. We believe that the main figures as they stand illustrate this to a suitable degree, while the both the high resolution supplemental figures and the extensive interactive resource at the web linked provided allow access to detail within of the data which is not commonly available.

1.2 “Additionally, example genes and example gene strips for ChIP-seq data need to be included in the primary figures to provide the reader examples genes to understand the patterns.”

We have included a full set of gene strips illustrating the binding patterns and associated modifications for each of the patterns/analyses discussed in the manuscript in new Supplemental Figure 6.

We have avoided including additional gene strips in the main figures to limit somewhat the overall complexity of figures and because we gave preference to the reviewer’s next comment (new Figure 2), and to the inclusion of the new analyses related to IRF4 at day-3 (new Figure 8) as discussed above.

1.3 “Without boiling complex patterns and concepts down to an example gene or loci readers will be lost. Consider moving Supplemental Fig 3 into Fig 1 and replacing one or more of the panels if necessary.”

We have followed this suggestion and moved Supplemental Figure 3 into the main Figures as new Figure 2. We have additionally added new gene tracks in particular for examples of the cell cycle gene patterns that follow release from CD40 stimulation. This figure is discussed on p15 line 274-292.

“2.1 No data is presented that allow the evaluation of the ChIP-seq data quality.

The inclusion of gene strips for loci to show transcription factor binding sites that fit each module and the histone patterns would help to evaluate the specificity of enrichment.’

We have given significant consideration to the reviewer’s point and have included examples of all patterns discussed in new Supplemental Figure 6. We believe that this allows a clear assessment of the quality of the ChIP-seq data.

We would also argue that the de novo motif detection and enrichments of highly specific motifs for each TF provides orthogonal verification across the full set of peaks for the specificity of the TF ChIPs. The activating histone mark data sets as shown in the deep tools analysis and in the new gene strips are also of high quality and follow patterns of distribution including skewing and displacement around active regulatory regions as expected for these factors.

“2.2 One concern is the enrichment in Fig 4 of the repressive histone marks H3K27me3 and H3K9me2 in all modules along with the other activating marks. To me this suggests these ChIPs did not work and are not specific. Modules U.k1-4 all have both activating and repressive marks and this needs to be reconciled to be classified as active gene clusters.

3.1 Were any ChIP controls used during the ChIP for H3K9me2 in samples that were treated with the G9a inhibitor? The H3K9me2 data in Fig 6 does not look specific with high background and it is known that in conditions with reduced antibody targets that background enrichment can be increased. The use of spike-in standards that several companies sell is a good workaround for some of these antibody issues. As mentioned above, Fig 6C shows enrichment of H3K9me2 at regions that also have activating marks. This is a major concern along with the potential low signal to noise observed in the gene strips.”

As discussed above we have taken this point on board. We originally considered that the indication that the G9A inhibitor treatment specifically depleted the H3K9me2 signal was evidence for

specificity. But we agreed with the reviewer's concern regarding the presence of repressive marks in the otherwise active clusters. We therefore performed a complete re-analysis excluding the repressive histone mark data, as we were not in a position to provide the technical controls or to repeat the ChIP-seq data. This reanalysis shows that the information content in the repressive histone mark data was indeed very low to negligible in terms of determining the K-means cluster types (the cluster patterns in the new figures are essentially unchanged after exclusion of this data). We therefore took the decision to exclude these data from the analysis and have reworked all elements of the manuscript that follow from this and all the figures. This appears as new Figure 4-7 and new Supplemental Figures 7-9 and related sections in the text on p20-25, although the overall associations are largely unaffected by the exclusion of the repressive marks. While this means we cannot make a statement regarding the impact of G9A inhibition on H3K9me2 state, it does not fundamentally detract from the manuscript, which instead takes a slightly different angle following the suggestion of reviewer 1 and data in new Figure 8.

4.1 *"The primary methods (not supplement) should state how many biological replicates were used for all experiments. For example, there are no biological replicates for the ChIP-seq data. While going back to repeat every ChIP is not necessary this should be clearly stated."*

This is now directly addressed in the methods, and we have included additional data to support conclusions as included in new Figure 8. This appears on p8 line 177-178 and p11 line 199-205 of the methods.

4.2 *"Additionally, the potential that false-negative results in the data, potentially from low ChIP quality as mentioned above, could explain some of the failure to identify some of the genes regulated by Blimp-1 in mouse plasmablasts."*

While we cannot exclude the contribution of false negatives we would argue that differing thresholds and approaches to peak detection are a more likely source than poor ChIP quality. As shown by the de novo motif detection data the enrichment of specific BLIMP1 motifs is highly significant and at a high percentage of peak sites. Furthermore, the selection of binding sites and overlap with IRF4 are also fully consistent with what we and others have previously demonstrated as the specific properties of these factors. We would also respectfully point out that the BLIMP1 ChIP in mouse ASCs has been generated with a bio-tag approach (Tellier et al. 2016). This clearly represent a substantial difference in approach and are not without issues. We also accept that differences in the differentiation state may contribute to the discrepancies and have addressed this in the discussion. We have provided a balanced assessment of this in the discussion on p30 line 654-665.

Minor Concerns:

1. *Consider labeling motifs in Fig 3C to easily let the readers know which match the known IRF4 composite motifs (AICE, EICE, ISRE).*

We have done so in new Figure 4.

2. *Improve the overall clarity of figures so a high-resolution supplemental figure is needed to see fine details of specific modules.*

We have tried to provide the best quality figures possible, but details of specific modules are not readily resolvable on main figure, which served more to illustrate overall patterns. We believe that the interested reader will find the fully interactive web-based resource and high-res supplemental figures a valuable addition to allow such detailed analysis.

July 28, 2020

RE: Life Science Alliance Manuscript #LSA-2020-00654-TR

Prof. Reuben M Tooze
University of Leeds
Leeds Institute of Medical Research
St James's University Hospital
Beckett Street
Leeds LS9 7TF
United Kingdom

Dear Dr. Tooze,

Thank you for submitting your revised manuscript entitled "A dichotomy of gene regulatory associations during the activated B-cell to plasmablast transition". Your manuscript was re-reviewed by one of the original referees whose report is attached below. We would be happy to publish your paper in Life Science Alliance pending final revisions necessary to meet our formatting guidelines.

-Please revise the discussion of the findings according to the remaining minor comments of referee #2

-please take a look at our Manuscript Preparation guidelines and order your manuscript sections accordingly

-please add the author contributions and a conflict of interest statement to the main manuscript

-please add a callout for Supp. Fig. S3B

-please list 10 authors et al. in the references

-Please increase font size in Figure panels 7D, 8D, S2, S5, S6 to ensure legibility

-please add scale bars to Fig. S3D & Fig. S10F

A. FINAL FILES:

-- High-resolution figure, supplementary figure and video files uploaded as individual files: See our

detailed guidelines for preparing your production-ready images, <http://www.life-science-alliance.org/authors>

B. MANUSCRIPT ORGANIZATION AND FORMATTING:

Sincerely,

Reilly Lorenz
Editorial Office Life Science Alliance
Meyershofstr. 1
69117 Heidelberg, Germany
t +49 6221 8891 414
e contact@life-science-alliance.org

Reviewer #2 (Comments to the Authors (Required)):

This revised manuscript by Tooze and colleagues represents a substantial revision that takes into account both reviewers concerns. In particular the authors have included sufficient controls to allow assessment of data quality, new experiments, reanalyzed data, and the addition of online resources is commendable to maximize the utility of such large datasets. The G9A inhibition experiment is limited in scope without the ability to measure the changes in repressive histone marks but that section of the manuscript is adequately revised to reflect the analysis that can be done. Only one minor issue is to make it clear in the sentence starting line 383 that ChIP-seq was performed on memory B cell derived PBs as in the section above.

August 12, 2020

RE: Life Science Alliance Manuscript #LSA-2020-00654-TRR

Prof. Reuben M Tooze
University of Leeds
Leeds Institute of Medical Research
St James's University Hospital
Beckett Street
Leeds LS9 7TF
United Kingdom

Dear Dr. Tooze,

Thank you for submitting your Research Article entitled "A dichotomy of gene regulatory associations during the activated B-cell to plasmablast transition". It is a pleasure to let you know that your manuscript is now accepted for publication in Life Science Alliance. Congratulations on this interesting work.

*****IMPORTANT:** If you will be unreachable at any time, please provide us with the email address of an alternate author. Failure to respond to routine queries may lead to unavoidable delays in publication.*******

DISTRIBUTION OF MATERIALS:

Again, congratulations on a very nice paper. I hope you found the review process to be constructive and are pleased with how the manuscript was handled editorially. We look forward to future exciting

submissions from your lab.

Sincerely,

Shachi Bhatt, PhD
Executive Editor, Life Science Alliance
<https://www.life-science-alliance.org/>
@LSAjournal